# EphB2 receptor cell-autonomous forward signaling mediates auditory memory recall and learning-driven spinogenesis

Asghar Talebian[1] & Mark Henkemeyer [1]*

While ephrin-B ligands and EphB receptors are expressed to high levels in the learning centers of the brain, it remains largely unknown how their trans-synaptic interactions contribute to memory. We find that EphB2 forward signaling is needed for contextual and sound-evoked memory recall and that constitutive over-activation of the receptor's intracellular tyrosine kinase domain results in enhanced memory. Loss of EphB2 expression does not affect the number of neurons activated following encoding, although a reduction of neurons activated after the sound-cued retrieval test was detected in the auditory cortex and hippocampal CA1. Further, spine density and maturation was reduced in the auditory cortex of mutants especially in the neurons that were dual-activated during both encoding and retrieval. Our data demonstrates that trans-synaptic ephrin-B-EphB2 interactions and forward signaling facilitate neural activation and structural plasticity in learning-associated neurons involved in the generation of memories.

[1] Department of Neuroscience and Kent Waldrep Center for Basic Research on Nerve Growth and Regeneration, University of Texas Southwestern Medical Center, Dallas, TX 75390, USA. *email: mark.henkemeyer@utsouthwestern.edu

Learning and memory is a highly complex process through which external stimuli and experiences are received, consolidated, stored, and recalled using an elaborate integration of neuronal circuits and several regions of the brain[1–5]. The hippocampus for instance forms a neural network important for contextual and episodic memories and is widely studied to understand the electrophysiological, cellular, and molecular basis of the synapse. Plasticity is an essential feature of learning and memory in which synapse strength is modulated through a highly dynamic process and regulated by several parameters. These include the presynaptic release of neurotransmitters and activation of postsynaptic neurons through an interplay of receptors and ion channels, leading to stimulation of intracellular signaling cascades, expression of immediate early genes (IEGs), and long-term potentiation (LTP), an electrophysiological measurement of persistent strengthening of synapses based on recent patterns of neural activity. Plasticity also involves more long-term cellular modifications in the connections between neurons, which results in structural changes to the synapses[6–8]. This structural plasticity can be observed as changes in the formation, destruction, or maturation shape of the spines that decorate the dendrites of neurons and form the major post-synaptic location of excitatory synapses. To better understand neural plasticity and memory formation, it is crucial to study the roles of specific molecules as an animal responds to an experience-induced learning paradigm.

The trans-synaptic ephrin-B ligands and cognate EphB receptor tyrosine kinases are conserved transmembrane proteins expressed to very high levels in the hippocampus and cortex[9]. Their interactions propagate bidirectional signals upon cell–cell contact that have been implicated in excitatory synaptic development, function, and plasticity[10–22] (reviewed in refs. [23–27]). Cell-autonomous forward signals mediated by intracellular components of EphB receptors, like the tyrosine kinase catalytic domain and C-terminal PDZ domain-binding motif, are thought to be particularly important in excitatory synapses. For instance, ephrin-B3 stimulated EphB2 forward signaling in the amygdala has been implicated in formation of innate fear responses by aiding the maturation of glutamatergic neurons[28,29]. Further, in addition to innate behaviors, studies of ephrin/Eph gene targeted mice using the Morris water maze[30] and classic fear conditioning (FC)[31–33] indicate the molecules encoded by these genes may participate in learning and memory[11,21,34–38]. However, it remains unknown whether ephrin-B-EphB interactions contribute to experience-driven neuron activation or if their signaling can affect the growth and/or remodeling of spines and synapses following exposure to a specific behavioral learning task that leads to formation of a particular memory.

Here, we show that cell-autonomous forward signaling mediated by the EphB2 receptor, but not the highly related EphB1 receptor, is necessary for generation of both contextual and sound-cued evoked memories induced by FC. We find that while EphB2$^{-/-}$ mutants exhibit a normal level of neurons that become activated during the encoding/training stage, they show reduced numbers of activated neurons after the sound-cued retrieval/recall test. Reductions in the complexity of dendritic spines were also detected in the EphB2$^{-/-}$ mutants, specifically affecting neurons within the auditory cortex associated with the learning/memory engram. Our data indicate that trans-synaptic ephrin-B-EphB2 interactions and forward signaling facilitate the expression of IEGs and modulate the structural plasticity of spines specifically within neurons associated with experience-driven memories, thus providing a trans-synaptic signaling mechanism that controls neuronal activation and morphological changes involved in learned behavior.

## Results

**EphB2-mutant mice have poor memory.** To assess potential roles for the EphB1 and EphB2 receptor tyrosine kinases in learned behavior we subjected gene targeted mutant mice to a sound-cued FC protocol. In this paradigm, a mouse has to learn during the encoding/training stage that the conditioned stimulus (CS, context/sound) will predict an unconditioned stimulus (US, foot shock) during the retrieval/testing stages. The following mutant mice used in this study were all generated in the Henkemeyer laboratory (see Methods): EphB1$^{-/-}$ protein-null[16], EphB1$^{lacZ/lacZ}$ C-terminal intracellular truncated[39], EphB2$^{-/-}$ protein-null[40], EphB2$^{lacZ/lacZ}$ C-terminal intracellular truncated[40], and EphB2$^{K661R/K661R}$ kinase-dead[41], EphB2$^{F620D/F620D}$ kinase-overactive[42], and EphB2$^{\Delta VEV/\Delta VEV}$ PDZ domain binding-dead intracellular point mutants[41]. The EphB homozygotes are all healthy, long-lived animals that exhibit relatively normal hearing, vision, and perception of acute pain[43–45].

We first conducted a control experiment on WT mice from our colony to make sure they learned as expected that the paired CS-US protocol would train them to predict a shock was coming when presented to either the same context on day 2 or the sound-cue (in a novel context) on day 4. Learning and memory was measured by an increase in time the CS-US mice spent freezing compared to mice that only experienced the CS. The CS-US mice exhibited a significant increase in the percentage of time freezing compared to the CS mice in both the context test and post-tone sound-cued test (Fig. 1a). This indicates our FC conditions result in formation of context recall and tone recall memories and that there is minimal effect from the general handling of mice during the procedures.

The ability of WT and EphB2-mutant mice to initially fear condition during the paired CS-US training experience was assessed by measuring time spent freezing for the 120″ period in the chamber before they were subjected to the sound and foot shock cycles (pre-tone) and then again for the 30–60″ periods immediatly following the first (CS-US#1, 60″), second (CS-US#2, 60″), and third (CS-US#3, 30″) sound-shock cycles (Fig. 1b). Statistical analysis of the data indicated that the percentage of freezing increased significantly post-tone compared to pre-tone in both WT and EphB2 mutant animals. The percentage of time freezing between pre-tone WT and pre-tone EphB2-mutant mice or between post-tone WT and post-tone EphB2-mutant mice were not significantly different. This indicates both WT and EphB2 mutant mice fear conditioned as a result of the CS-US paradigm employed and that there were no significant differences between the two genotype classes with regards to their ability to become trained during the encoding step.

The percentage of time freezing observed for the various EphB-mutant mice subjected to the contextual test (Fig. 1c) and sound-cued test (Fig. 1d) are shown, and isolated male/female data are also provided (Supplementary Fig. 1). The EphB2$^{-/-}$ knockout mice exhibited highly significant reductions in time spent freezing compared to WT controls in tests for both context recall and sound-cued recall. The intracellular truncated EphB2$^{lacZ/lacZ}$-mutant mice also exhibited significant reductions in both tests compared to the WT mice. This data indicate a need for EphB2 in the formation of both context recall and tone recall memories and that an intact intracellular domain is important.

EphB1$^{-/-}$ knockout and EphB1$^{lacZ/lacZ}$ intracellular truncated mutant mice exhibited no significant differences in freezing compared to WT mice in either the contextual or sound-cued tests (Fig. 1c, d). EphB1$^{-/-}$;EphB2$^{-/-}$ double knockouts, however, did exhibit highly significant decreases in time freezing compared to WT in both tests. The results with the double mutants are quite similar to those observed for the EphB2$^{-/-}$ and EphB2$^{lacZ/lacZ}$ single mutants, solidifying the idea that while

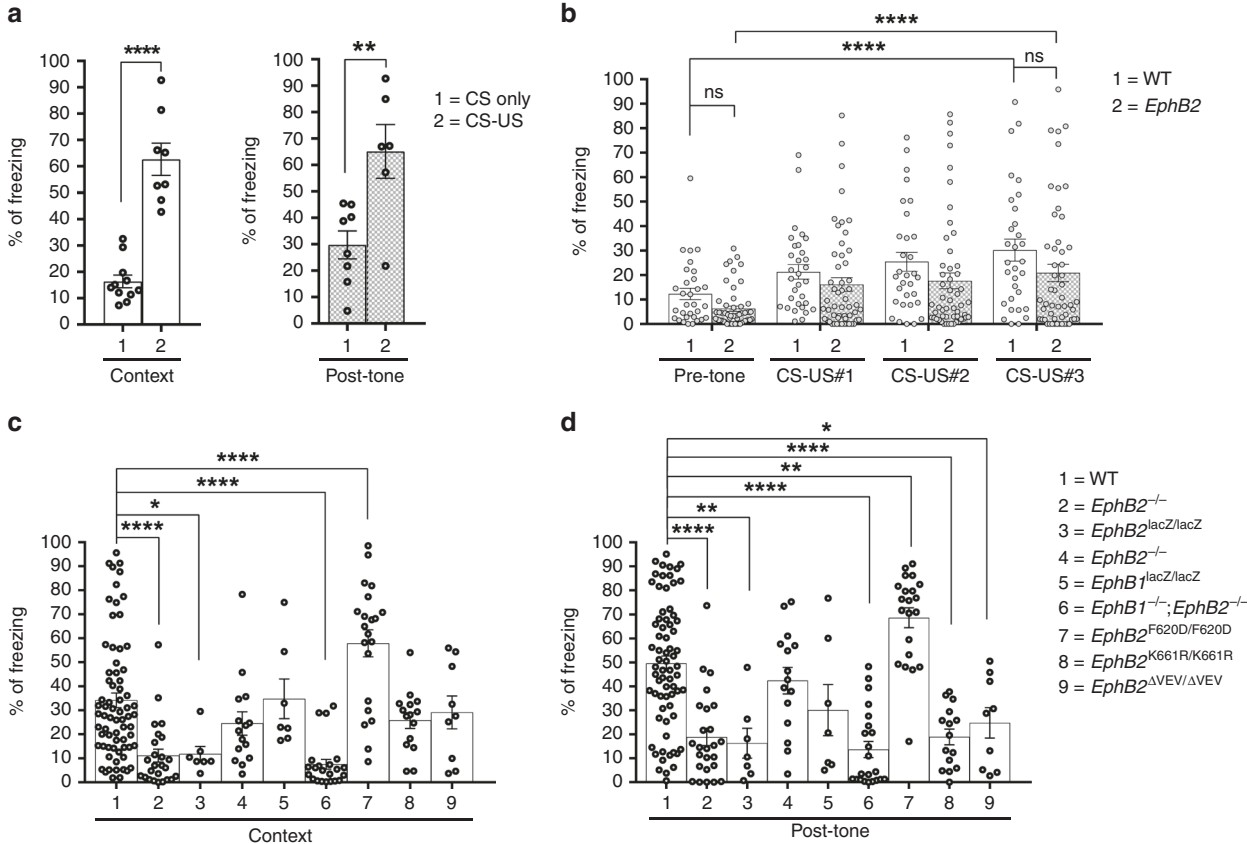

**Fig. 1** *EphB*-mutant mice exhibit reduced contextual and sound-cued FC memory. Following encoding/training (day 0), mice were subjected to the context test (day 2) and the sound-cued test (day 4). The percentage of time each mouse spent freezing is shown. **a** Freezing in a control group of WT mice that were either subjected to CS only training or CS-US training. For contextual test, CS only group 16.36% ± 2.42, $n = 11$; CS-US group 62.65% ± 6.1, $n = 8$; unpaired *t*-test, Welch-corrected $t_{(9.21)} = 7.04$; for post-tone test, CS only group 29.79% ± 5.29, $n = 8$; CS-US group 65.14% ± 10.17, $n = 6$; unpaired *t*-test, $t_{(12)} = 3.32$. **b** Learning during FC training was determined by comparing time freezing during the 120″ prior to the first sound-shock (pre-tone) to the time freezing during the 30–60″ periods immediately following the first (CS-US#1, 60″), second (CS-US#2, 60″), and third (CS-US#3, 30″) sound-shock cycles. $n = 31$ WT, $n = 53$ a pooled collection of $EphB2^{-/-}$, $EphB2^{lacZ/lacZ}$, and $EphB1^{-/-};EphB2^{-/-}$ mutants; two-way repeated measures ANOVA, interaction $F_{(3,246)} = 0.29$, $p = 0.8326$; effect of genotype $F_{(1,82)} = 4.27$, $p = 0.0419$; effect of CS-US $F_{(3,246)} = 15.8$, $p < 0.0001$; Sidak multiple comparison tests; effect of genotype pre-tone in WT vs. mutant $t_{(328)} = 1.32$; effect of genotype CS-US#3 in WT vs. mutant $t_{(328)} = 2.06$; effect of CS-US pre-tone vs. CS-US#3 in WT $t_{(246)} = 4.59$; effect of CS-US pre-tone vs. CS-US#3 in mutant $t_{(246)} = 4.88$. **c, d** Time freezing for each mouse in the context test (**c**) and sound-cued test (**d**). $n = 70$ WT, $\underline{n} = 25$ $EphB2^{-/-}$ knockout, $n = 7$ $EphB2^{lacZ/lacZ}$ intracellular-truncated, $n = 15$ $EphB1^{-/-}$ knockout, $n = 7$ $EphB1^{lacZ/lacZ}$ intracellular-truncated, $n = 22$ $EphB1^{-/-};EphB2^{-/-}$ double knockout, $n = 21$ $EphB2^{F620D/F620D}$ kinase-overactive, $n = 15$ $EphB2^{K661R/K661R}$ kinase-dead, and $n = 9$ $EphB2^{\Delta VEV/\Delta VEV}$ PDZ domain binding-dead. For contextual test (**c**), one-way ANOVA, treatment $F_{(8,182)} = 11.76$, $p < 0.0001$; Dunnett multiple comparison test to WT; $EphB2^{-/-}$ $q_{(93)} = 4.79$; $EphB2^{lacZ/lacZ}$ $q_{(75)} = 2.73$; $EphB1^{-/-}$ $q_{(83)} = 1.64$; $EphB1^{lacZ/lacZ}$ $q_{(75)} = 0.078$; $EphB1^{-/-};EphB2^{-/-}$ $q_{(90)} = 5.27$; $EphB2^{F620D/F620D}$ $q_{(89)} = 4.61$; $EphB2^{K661R/K661R}$ $q_{(83)} = 1.43$; $EphB2^{\Delta VEV/\Delta VEV}$ $q_{(77)} = 0.69$. For post-tone sound-cued test (**d**), one-way ANOVA, treatment $F_{(8,182)} = 16.43$, $p < 0.0001$; Dunnett multiple comparison test to WT; $EphB2^{-/-}$ $q_{(93)} = 6.06$; $EphB2^{lacZ/lacZ}$ $q_{(75)} = 3.86$; $EphB1^{-/-}$ $q_{(83)} = 1.16$; $EphB1^{lacZ/lacZ}$ $q_{(75)} = 2.26$; in $EphB1^{-/-};EphB2^{-/-}$ $q_{(90)} = 6.76$; $EphB2^{F620D/F620D}$ $q_{(89)} = 3.51$; $EphB2^{K661R/K661R}$ $q_{(83)} = 4.95$; $EphB2^{\Delta VEV/\Delta VEV}$ $q_{(77)} = 3.22$. Error bars are standard error of the mean (SEM); ns non-significant; *$p < 0.05$, **$p < 0.01$, ***$p < 0.001$, ****$p < 0.0001$

EphB1 is dispensable for both forms of memory, the EphB2 receptor is essential.

Given the importance of the EphB2 receptor intracellular domain for context recall and tone recall memories, specific point mutant mice that affect its tyrosine kinase catalytic activity were tested. While $EphB2^{K661R/K661R}$ kinase-dead mutants exhibited normal freezing in the context test compared to WT (Fig. 1c), they showed a highly significant reduced performance in the sound-cued test (Fig. 1d). Interestingly, the $EphB2^{F620D/F620D}$ kinase-overactive mutants showed significant increases in freezing compared to the WT mice in both contextual and sound-cued tests, that may impact females more than males (Supplementary Fig. 1). Together, the K661R and F620D results indicate EphB2 tyrosine kinase activity plays an important role in sound-cued memory and that increases in catalytic activity above

normal leads to an enhancement in both contextual and sound-cued tests.

Potential roles for EphB2 interacting with PDZ domain-containing proteins were assessed in $EphB2^{\Delta VEV/\Delta VEV}$ mutants. Similar to the kinase-dead results, $EphB2^{\Delta VEV/\Delta VEV}$-mutant mice when compared to WT counterparts also exhibited normal freezing in the test for context recall test (Fig. 1c), but they performed poorly in the sound-cued tone recall test (Fig. 1d). This suggest EphB2 binding to PDZ domain proteins is also very important for sound-cued memory. Because the $EphB2^{lacZ/lacZ}$ mutation led to significant reduction in both context and sound-cued freezing, but the kinase-dead and PDZ binding-dead only affected sound-cued memory, it seems that either (1) tyrosine kinase and PDZ binding functions are redundant and that either one or the other is sufficient for contextual memory, or (2) that

something other than catalytic activity or PDZ interactions of EphB2 that is disrupted by the lacZ mutation is important.

**EphB2 in neuron activation following memory retrieval.** We sought to determine if loss of EphB2 receptor may affect the number of neurons activated in FC by assessing for induction of IEG expression after the encoding stage and then again 4 days later following the sound-cued memory retrieval test. To identify neurons activated during encoding, we utilized targeted recombination in active populations (TRAP) in which CreER$^{T2}$ is targeted into the *Fos* locus (*Fos*$^{Trap/+}$) and becomes transiently expressed in the subset of neurons that become activated[46,47]. Cells that express CreER$^{T2}$ undergo recombination only when tamoxifen is present, allowing genetic access to neurons that were active during a short time window of ~6 h after the active metabolite 4-hydroxytamoxifen (4-OHT) is injected. To visualize recombined neurons, Ai9 knockin allele of the *Rosa26* locus (*R26*$^{Ai9/+}$) was incorporated to provide high-level ubiquitous expression of the red fluorescent protein tdTomato (Tom) after excision of the loxP-flanked transcriptional stop signal[48]. To obtain animals for the study, *EphB2*$^{+/−}$;*Fos*$^{Trap/+}$;*R26*$^{Ai9/Ai9}$ mice were first generated and then mated to *EphB2*$^{+/−}$ mice to produce *EphB2*$^{−/−}$;*Fos*$^{Trap/+}$;*R26*$^{Ai9/+}$ mutant and *EphB2*$^{+/+}$; *Fos*$^{Trap/+}$;*R26*$^{Ai9/+}$ WT littermates. These animals were given a single dose of 4-OHT immediately prior to being subjected to sound-cued FC encoding. Four days after training, mice were subjected to the sound-cued recall test and after 90 min to allow for IEG expression the brains were collected, sectioned, and immunoreacted with anti-Fos specific antibodies. Importantly, no significant effect of tamoxifen was observed on contextual or sound-cued freezing in the WT or *EphB2*$^{−/−}$-mutant mice (Supplementary Fig. 2). Home cage control *EphB2*$^{+/+}$;*Fos*$^{Trap/+}$; *R26*$^{Ai9/+}$ mice that were injected with 4-OHT but not subjected to FC encoding or retrieval procedures were also included in the analysis. Based on IEG expression readout, Trapped neurons that were activated during a short window of time following exposure to sound-cued FC encoding session will be indelibly labeled with Tom$^+$ fluorescence and then neurons activated 4 days later after the sound-cued recall test will be identified by anti-Fos immunofluorescence. Our goal was to capture, quantify, and study the learning/memory-associated neurons as those which were Tom$^+$/Fos$^+$ double-positive and activated both during encoding and retrieval stages, Tom$^+$ single-positive cells activated only during encoding, and Fos$^+$ single-positive cells activated only during retrieval in order to identify for any abnormalities in the *EphB2*$^{−/−}$ mutants compared to their WT littermates.

We focused analysis on multiple regions of the brain implicated in FC, including the dentate gyrus (Fig. 2), CA1 and CA3 regions of the hippocampus (Fig. 3), the auditory cortex (Fig. 4), and the cortical, central, and basolateral regions of the amygdala (Fig. 5). Comparing the WT mice subjected to FC to their corresponding WT home cage control mice (WT-h), no significant changes in the numbers of Tom$^+$/Fos$^+$ double-positive learning-associated neurons activated both during encoding and retrieval were observed in the hippocampus (Figs. 2 and 3). However, significant increases in Tom$^+$/Fos$^+$ double-positive learning-associated neurons were observed in the trained WT mice compared to WT home cage mice specifically in the auditory cortex (Fig. 4) and cortical amygdala (Fig. 5). The auditory cortex of the trained WT mice further showed a significant increase in the number of Tom$^+$ single-positive neurons activated only during encoding compared to the WT home cage (Fig. 4). Furthermore, all regions of the brain assessed showed significant increases in the numbers of Fos$^+$ single-positive cells activated exclusively following retrieval in WT

trained mice compared to the WT home cage mice (Figs. 2–5 and Table 1). The data are consistent with other reports that show *Fos*$^{Trap/+}$ is readily able to Trap neurons in the auditory cortex and amygdala, though it is less efficient in certain other brain regions including the hippocampus[46,47].

When the trained *EphB2*$^{−/−}$ mutants were compared to their trained WT littermate counterparts, no significant differences in the numbers of Tom$^+$/Fos$^+$ double-positive or Tom$^+$ single-positive neurons were noted for any region of the brain assessed (Figs. 2–5 and Table 1). This suggests loss of EphB2 does not affect the number of neurons that become activated following FC encoding and is consistent with the ability of *EphB2*$^{−/−}$ mutants to initially fear condition like their WT littermates during encoding/training stage (Fig. 1b). However, the *EphB2*$^{−/−}$-mutant brains exhibited significant decreases in Fos$^+$ single-positive neurons compared to WT counterparts in the hippocampal CA1 region (Fig. 3) and in the auditory cortex (Fig. 4). This data indicate loss of EphB2 leads to a reduction in the number of neurons that become activated following the sound-cued retrieval test and is consistent with a reduction in the percentage of time freezing during this stage (Fig. 1d).

**EphB2 in dendritic complexity of learning-associated neurons.** We next assessed whether EphB2 may participate in morphological changes associated with neurons that are involved in formation of a particular memory. To accomplish this, the Thy1-GFP$^M$ reporter transgene[49] was incorporated into the cross that generated the *EphB2*$^{−/−}$;*Fos*$^{Trap/+}$;*R26*$^{Ai9/+}$ and *EphB2*$^{+/+}$; *Fos*$^{Trap/+}$;*R26*$^{Ai9/+}$ trapping mice. Because Thy1-GFP$^M$ brightly labels a small random number of excitatory neurons in the cortex and hippocampus, it is a useful tool to assess dendritic complexity and spine density/morphology of individual neurons[22,28,29]. We aimed to score Thy1-GFP$^M$-labeled neurons that became activated during FC encoding and/or were activated after the recall test, anticipating that loss of EphB2 in such Tom$^+$/Fos$^+$/GFP$^+$ dual-activated neurons and Tom$^−$/Fos$^+$/GFP$^+$ recall-activated neurons will impact their abilities to remodel dendrites and spines upon FC. We focused on the hippocampal CA1 and auditory cortex as the above data indicated these are the two regions with significant reductions in Fos$^+$ recall-activated neurons in the *EphB2*$^{−/−}$ mutants.

Dendritic complexity was scored using Sholl analysis. In the CA1 region we were able to score Tom$^−$/Fos$^+$/GFP$^+$ recall-activated neurons and compare to Tom$^−$/Fos$^−$/GFP$^+$ dual-negative 'unlearned' counterparts, though no significant differences between *EphB2*$^{−/−}$ mutants and WT was noted (Fig. 6a). Due to poor trapping efficiency, very few Tom$^+$/Fos$^+$/GFP$^+$ dual-activated neurons were detected in the CA1 and this prevented analysis with Thy1-GFP$^M$. All three classes of neurons, however, were easily identified in the auditory cortex and were subjected to Sholl analysis (Fig. 7). The data show Tom$^+$/Fos$^+$/GFP$^+$ dual-activated neurons in the auditory cortex of *EphB2*$^{−/−}$ mutants exhibited a decrease in dendritic complexity compared to WT counterparts as evident by a significantly reduced average number of intersections at radius 25 and 40 μm (Fig. 7a). We then compared the average number of intersections between Tom$^+$/Fos$^+$/GFP$^+$ neurons to unlearned neurons (Tom$^−$/Fos$^−$/GFP$^+$) in WT brains (Fig. 7b) separately from *EphB2*$^{−/−}$-mutant brains (Fig. 7c). Although a mild increase was observed in the number of intersections in WT, no significant changes were noted. Further, no difference in dendritic complexity was noted in the Tom$^−$/Fos$^+$/GFP$^+$ recall-activated neurons from the Tom$^−$/Fos$^−$/GFP$^+$ dual-negative compared to the 'unlearned' counterparts in the *EphB2*$^{−/−}$ mutants.

Consistent with the Sholl analysis, the total length of dendritic branches was significantly reduced in *EphB2*$^{−/−}$

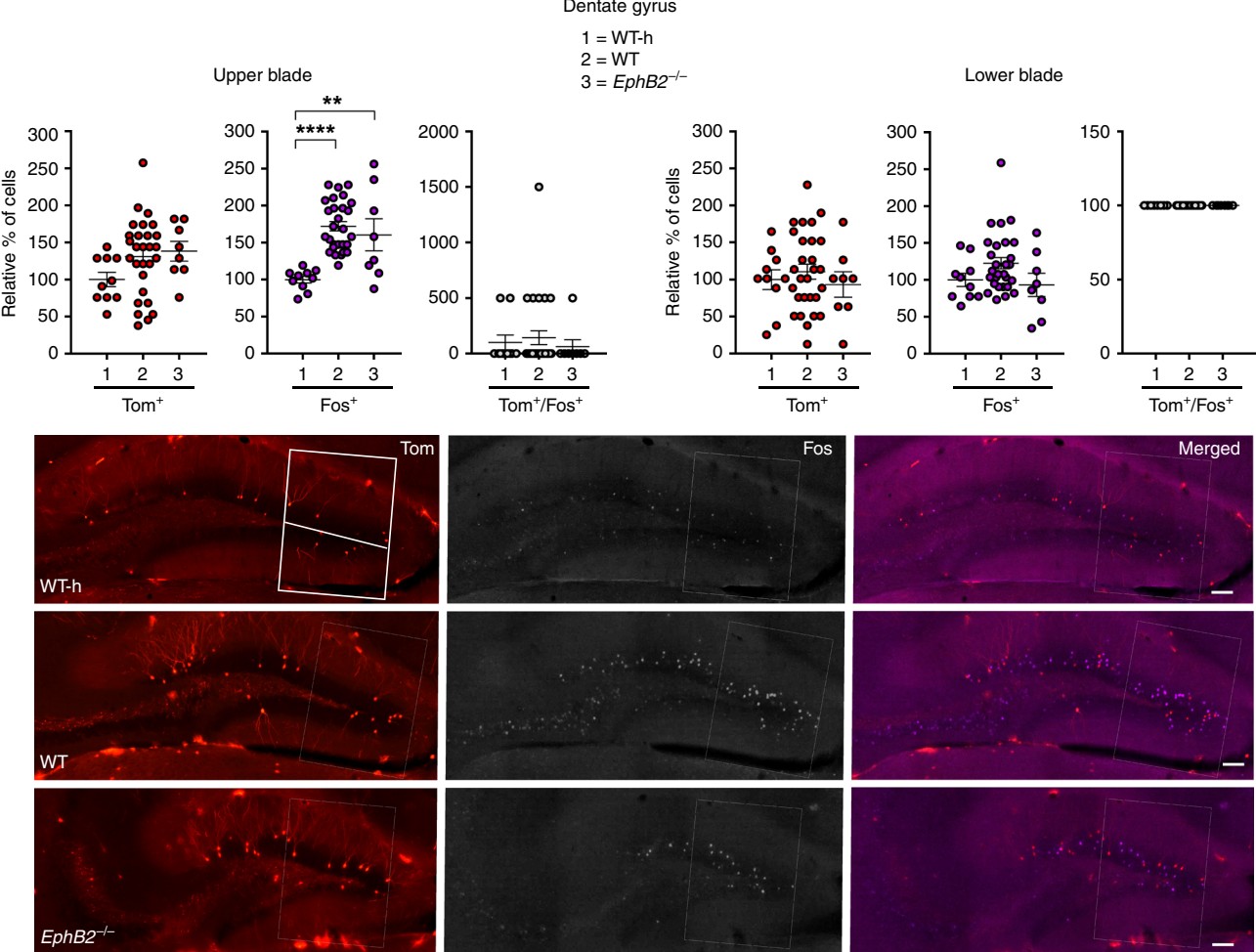

**Fig. 2** Analysis of Tom+ and Fos+ learning-associated neurons in the dentate gyrus (DG) of *EphB2* mutant mice. Neurons activated exclusively following encoding (Tom+ single-positive), exclusively following sound-cued retrieval (Fos+ single-positive), and following both encoding and retrieval (Tom+/Fos+ dual-positive) were imaged and counted in defined upper and lower blades of the DG from *EphB2−/−* mutants and WT littermates subjected to FC and WT home cage controls not subjected to FC (WT-h). Scatter plots show the number of Tom+ single-positive neurons, Fos+ single-positive neurons, and Tom+/Fos+ dual-positive neurons. Representative confocal images of Tom+ and Fos+ labeled neurons in the dentate gyrus are shown with the boxes indicating quantification areas. $n = 10$ hemisphere WT-h, $n = 28$ hemisphere WT, and $n = 8$ hemisphere *EphB2−/−*. For Fos+ single-positive neurons in DG upper blade, one-way ANOVA, $F_{(2,43)} = 14.17$, $p < 0.0001$; Tukey multiple comparison test; WT-h vs WT $q_{(36)} = 7.49$, $p < 0.0001$; WT-h vs *EphB2−/−* $q_{(16)} = 4.89$, $p = 0.0035$; WT vs *EphB2−/−* $q_{(34)} = 1.102$, $p = 0.7171$. Error bars are standard error of the mean (SEM); $**p < 0.01$, $****p < 0.0001$. Scale bar $= 100$ μm

mutant Tom+/Fos+ dual-activated neurons in the auditory cortex compared to the WT counterparts (Fig. 7d), but not in Tom−/Fos+/GFP+ recall-activated neurons or Tom−/Fos−/GFP+ dual-negative 'unlearned' counterparts. Thus, loss of EphB2 affects dendritogenesis in the select group of Tom+/Fos+ dual-activated, learning-associated neurons following FC.

**EphB2 in spinogenesis of learning-associated neurons**. The Thy1-GFP^M reporter was also used to investigate for learning-induced morphological changes in spines which are the post-synaptic structures that extend from dendrites and comprise the bulk of glutamatergic synapses. The assessment of spines (i.e. numbers, length, size/shape) of neurons in the CA1 and auditory cortex was used to determine if there are changes in density and/ or proportion of mushroom shaped, stubby, and thin spines[22,28,29].

In the CA1 region we were able to score spine morphology in Tom−/Fos+/GFP+ recall-activated neurons and compare to Tom−/Fos−/GFP+ dual-negative 'unlearned' counterparts,

though no significant differences between *EphB2−/−* mutants and WT was noted (Fig. 6b).

Analysis of spine morphologies in the auditory cortex was striking (Fig. 8a). While the Tom−/Fos-/GFP+ 'unlearned' neurons showed no difference between the WT and *EphB2−/−* mutants, the Tom+/Fos+/GFP+ dual-activated neurons from the knockouts showed highly significant reductions in total spine density (t) that affected the thin (T) and especially the more mature mushroom (M) shaped spines (Fig. 8b). Further, the Tom−/Fos+/GFP+ recall-activated neurons in *EphB2−/−* mutants also exhibited a significant reduction in total spine density that was reflected by a specific and highly significant reduction in mushroom shaped spines (Fig. 8b).

We further compared spine density and morphology in Tom+/Fos+/Thy1+ dual-activated 'learning-associated' neurons to the Tom−/Fos−/GFP+ 'unlearned' neurons from the same genotype (Fig. 8c). In brains from WT mice, the density of total spines was significantly increased in the learning-associated neurons compared to unlearned neurons, and this increase was noted in

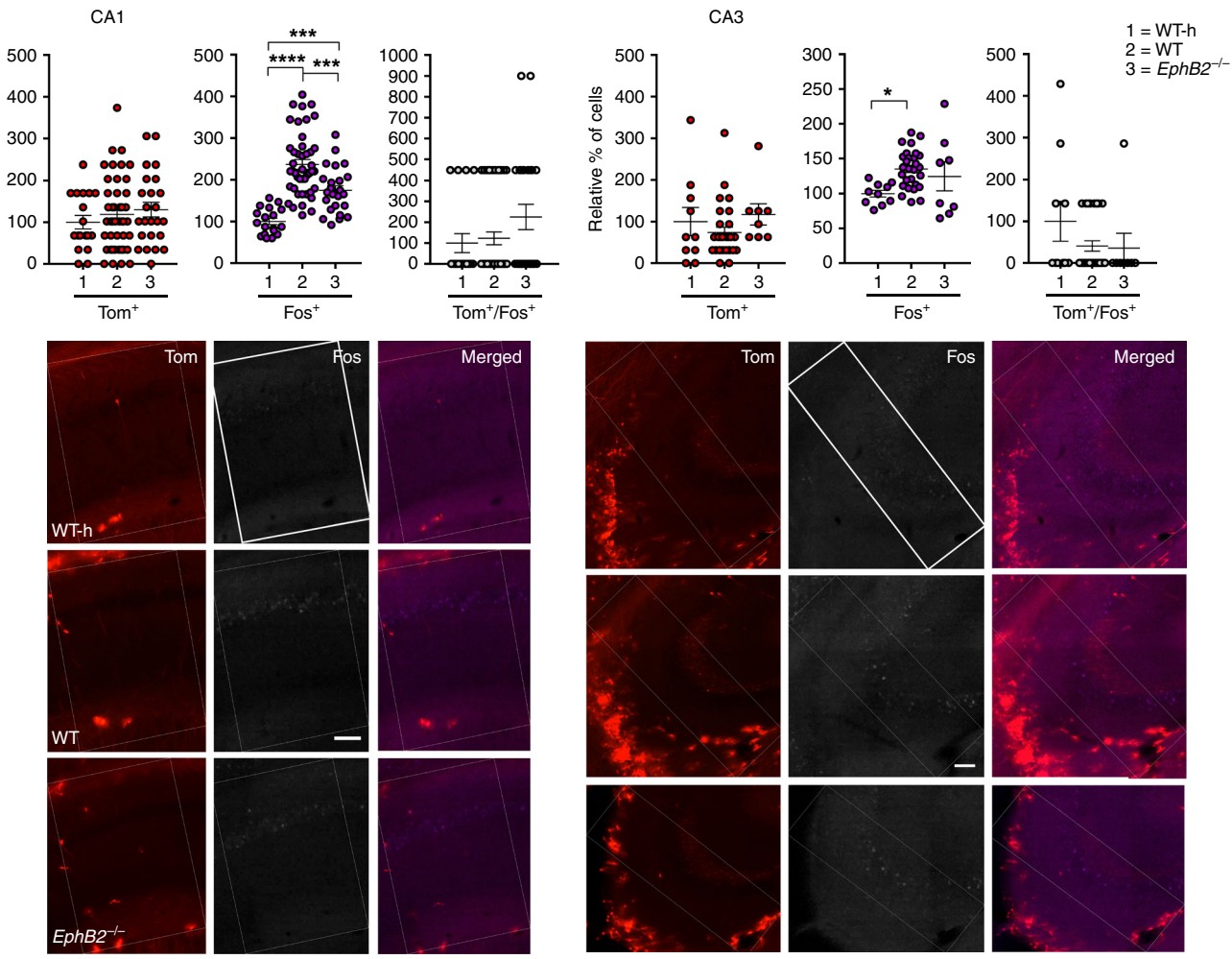

**Fig. 3** Analysis of Tom$^+$ and Fos$^+$ learning-associated neurons in the hippocampus of *EphB2* mutant mice. Neurons activated exclusively following encoding (Tom$^+$ single-positive), exclusively following sound-cued retrieval (Fos$^+$ single-positive), and following both encoding and retrieval (Tom$^+$/Fos$^+$ dual-positive) were imaged and counted in the CA1 and CA3 regions from *EphB2$^{-/-}$* mutants and WT littermates subjected to FC and WT home cage controls not subjected to FC (WT-h). Scatter plots show the number of Tom$^+$ single-positive neurons, Fos$^+$ single-positive neurons, and Tom$^+$/Fos$^+$ dual-positive neurons. Representative confocal images of Tom$^+$ and Fos$^+$ labeled neurons in the CA1 and CA3 regions are shown with the boxes indicating quantification areas. For CA1: $n = 18$ hemisphere WT-h, $n = 44$ hemisphere WT, and $n = 24$ hemisphere *EphB2$^{-/-}$*; Fos$^+$ single-positive neurons, one-way ANOVA, $F_{(2,83)} = 30.52$, $p < 0.0001$; Tukey multiple comparison test; WT-h vs WT $q_{(60)} = 10.85$, $p < 0.0001$; WT-h vs *EphB2$^{-/-}$* $q_{(40)} = 5.30$, $p = 0.0010$; WT vs *EphB2$^{-/-}$* $q_{(66)} = 5.453$, $p = 0.0007$. For CA3: $n = 10$ hemisphere WT-h, $n = 28$ hemisphere WT, and $n = 8$ hemisphere *EphB2$^{-/-}$*; Fos$^+$ single-positive neurons, one-way ANOVA, $F_{(2,43)} = 4.31$, $p = 0.0197$; Tukey multiple comparison test; WT-h vs WT $q_{(36)} = 4.15$, $p = 0.0145$; WT-h vs *EphB2$^{-/-}$* $q_{(16)} = 2.24$, $p = 0.2629$; WT vs *EphB2$^{-/-}$* $q_{(34)} = 1.16$, $p = 0.6928$. Error bars are standard error of the mean (SEM); *$p < 0.05$, ***$p < 0.001$, ****$p < 0.0001$. Scale bar = 100 μm

thin shaped and mushroom shaped spines. Remarkably, in the *EphB2$^{-/-}$* brains, the Tom$^+$/Fos$^+$/Thy1$^+$ dual-activated neurons exhibited no differences in spine density or morphologies compared to their Tom$^-$/Fos$^-$/GFP$^+$ 'unlearned' counterparts. The data indicate that functional EphB2 receptor protein is necessary for the elaboration and maturation of additional new spines that are induced upon FC.

## Discussion

While much is known about the synaptic proteins that participate in neural transmission and plasticity in the brain, the identification of molecules and signaling pathways that directly participate in formation of a specific learned memory remains elusive. We show here that EphB2 receptor forward signaling is necessary for FC-induced learning and memory, with both its intracellular tyrosine kinase catalytic activity and ability to couple to PDZ domain containing proteins being particularly important for

sound-cued, hippocampal-independent memories. The role for EphB2 is strengthened by our finding that the F620D point mutation, which constitutively activates its tyrosine kinase domain leads to enhanced contextual and sound-cued memories, and is consistent with a recent report that shows optogenetic activation of this receptor also leads to increased auditory FC memory[38]. By indelibly labeling with dtTomato the neurons that become activated shortly after FC encoding and combining with analysis of Fos protein expression following the sound-cued retrieval test, we determined that loss of EphB2 leads to reduced numbers of activated cells in the CA1 region and auditory cortex, though apparently affecting only the Fos$^+$-single positive neurons activated exclusively during memory recall. Importantly, we find that Tom$^+$/Fos$^+$ learning-associated neurons in the auditory cortex, which become activated during both the encoding and retrieval stages show decreased dendritic complexity and fail to elaborate new spines or increase the numbers of mature spines in

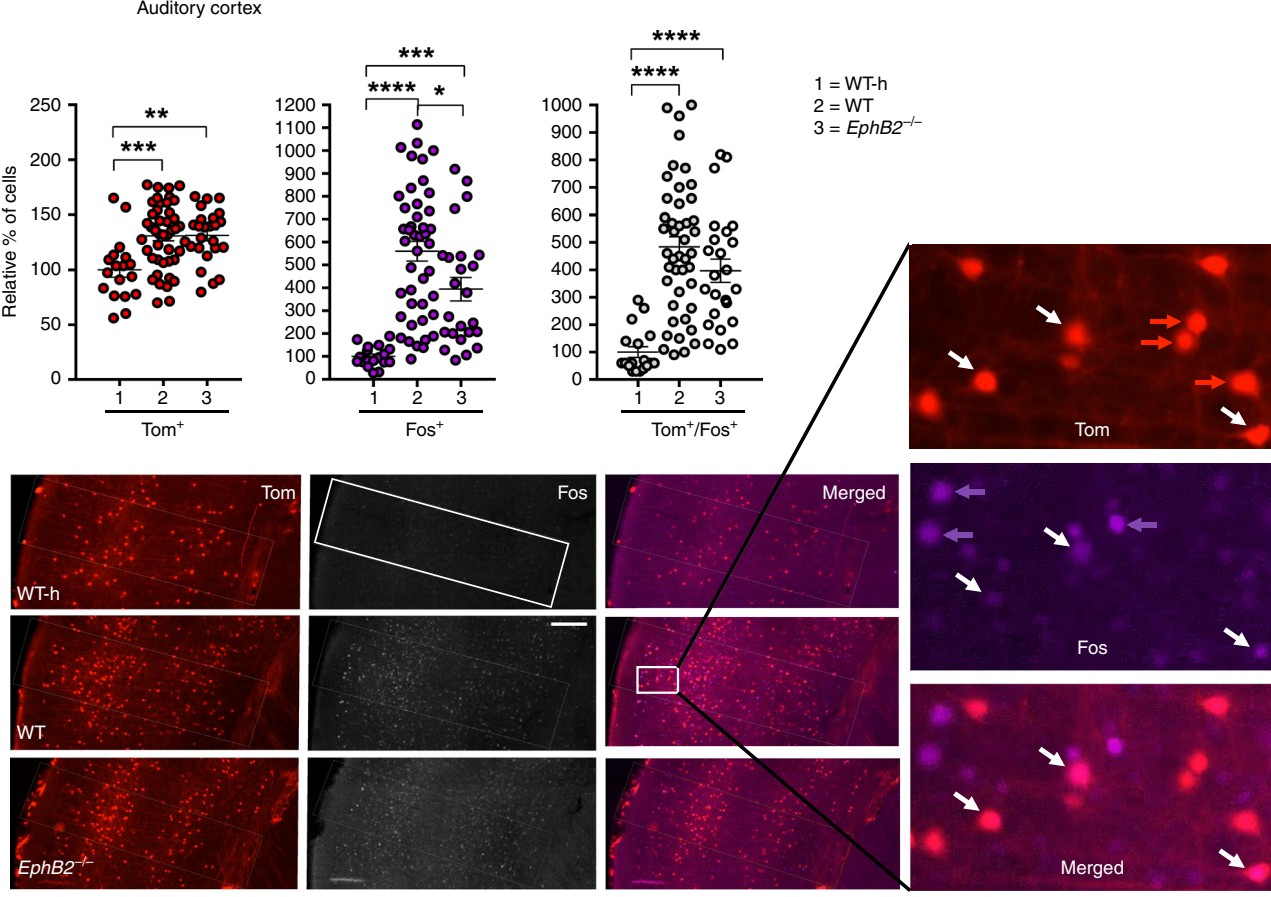

**Fig. 4** Analysis of Tom$^+$ and Fos$^+$ learning-associated neurons in the auditory cortex of *EphB2* mutant mice. Neurons activated exclusively following encoding (Tom$^+$ single-positive), exclusively following sound-cued retrieval (Fos$^+$ single-positive), and following both encoding and retrieval (Tom$^+$/Fos$^+$ dual-positive) were imaged and counted in the auditory cortex from *EphB2$^{-/-}$* mutants and WT littermates subjected to FC and WT home cage controls not subjected to FC (WT-h). Scatter plots show the number of Tom$^+$ single-positive neurons, Fos$^+$ single-positive neurons, and Tom$^+$/Fos$^+$ dual-positive neurons. Representative confocal images of Tom$^+$ and Fos$^+$ labeled neurons in the auditory cortex are shown with the box indicating quantification area. A magnified image from the WT auditory cortex is shown indicating individual Tom$^+$-labeled cells (red arrows), Fos$^+$-labeled cells (purple arrows), and Tom$^+$/Fos$^+$-double-labeled cells (white arrows). $n = 18$ hemisphere WT-h, $n = 44$ hemisphere WT, and $n = 24$ hemisphere *EphB2$^{-/-}$*. For Tom$^+$-single positive neurons, one-way ANOVA, $F_{(2,83)} = 8.66$, $p = 0.0004$; Tukey multiple comparison test; WT-h vs WT $q_{(60)} = 5.55$, $p = 0.0005$; WT-h vs *EphB2$^{-/-}$* $q_{(40)} = 5.05$, $p = 0.0017$; WT vs *EphB2$^{-/-}$* $q_{(66)} = 0.09$, $p = 0.9979$. For Fos$^+$-single positive neurons, one-way ANOVA, $F_{(2,83)} = 22.61$, $p < 0.0001$; Tukey multiple comparison test; WT-h vs WT $q_{(60)} = 9.48$, $p < 0.0001$; WT-h vs *EphB2$^{-/-}$* $q_{(40)} = 5.44$, $p = 0.0007$; WT vs *EphB2$^{-/-}$* $q_{(66)} = 3.77$, $p = 0.0248$. For Tom$^+$/Fos$^+$-double positive neurons, one-way ANOVA, $F_{(2,83)} = 21.14$, $p < 0.0001$; Tukey multiple comparison test; WT-h vs WT $q_{(60)} = 9.17$, $p < 0.0001$; WT-h vs *EphB2$^{-/-}$* $q_{(40)} = 6.36$, $p < 0.0001$; WT vs *EphB2$^{-/-}$* $q_{(66)} = 2.30$, $p = 0.2413$. Error bars are standard error of the mean (SEM); $*p < 0.05$, $**p < 0.01$, $***p < 0.001$, $****p < 0.0001$. Scale bar = 100 μm

*EphB2$^{-/-}$*-mutant mice following encoding. This inability to elaborate additional mature spines also affects the Fos$^+$-single positive neurons that become activated only following the retrieval. As spine numbers and distribution types are not affected in the 'unlearned' double-negative neurons not associated with FC, it appears EphB2 mediates the structural plasticity and activation of neurons involved in formation of a specific learned memory.

Regarding the use of FosTrap to identify the neurons that become activated shortly after a specific FC training event, we realized that its usefulness to study the hippocampus is limited as trapping here was very inefficient resulting in few cells that became labeled with Tom$^+$. Thus, in our analysis of the hippocampus, the neurons identified as Fos$^+$-single positive may indeed also had been activated during encoding though they just didn't get trapped. Likewise, it is also possible that Tom$^+$ labeling in the auditory cortex and amygdala did not identify all cells activated during the encoding stage and that the cells we identified as Fos$^+$-single positive may have actually become activated

during both encoding and retrieval, but again were just not trapped. Early in our analysis we also tested the related Arc-Trap line[46], though here way too many cells became Tom$^+$ labeled, even in the absence of 4-OHT administration (not shown). A new mouse, Trap2, has recently been reported that is more active in the hippocampus and in other regions of the brain implicated in learning and memory[50,51]. This Trap2 mouse could be useful to further explore the roles of EphB2, particularly to assess neuronal activation and re-activation in contextual hippocampal-based memories as well as in other regions implicated in FC such as the prelimbic cortex. We were also unable to assess Thy1-GFP$^M$ labeling in the amygdala as too many GFP$^+$ cells were present to unequivocally identify whether a dendritic segment imaged stemmed from a Tom$^+$, Fos$^+$, Tom$^+$/Fos$^+$, or a Tom$^-$/Fos$^-$ neuron. Further, while the control data in Fig. 1a shows WT mice subjected to only CS treatment exhibit significantly lower freezing than the mice subjected to the CS-US training exercise, the brains from these mice were not included in the cellular analyses. Thus, the formal possibility exists, albeit unlikely, that structural

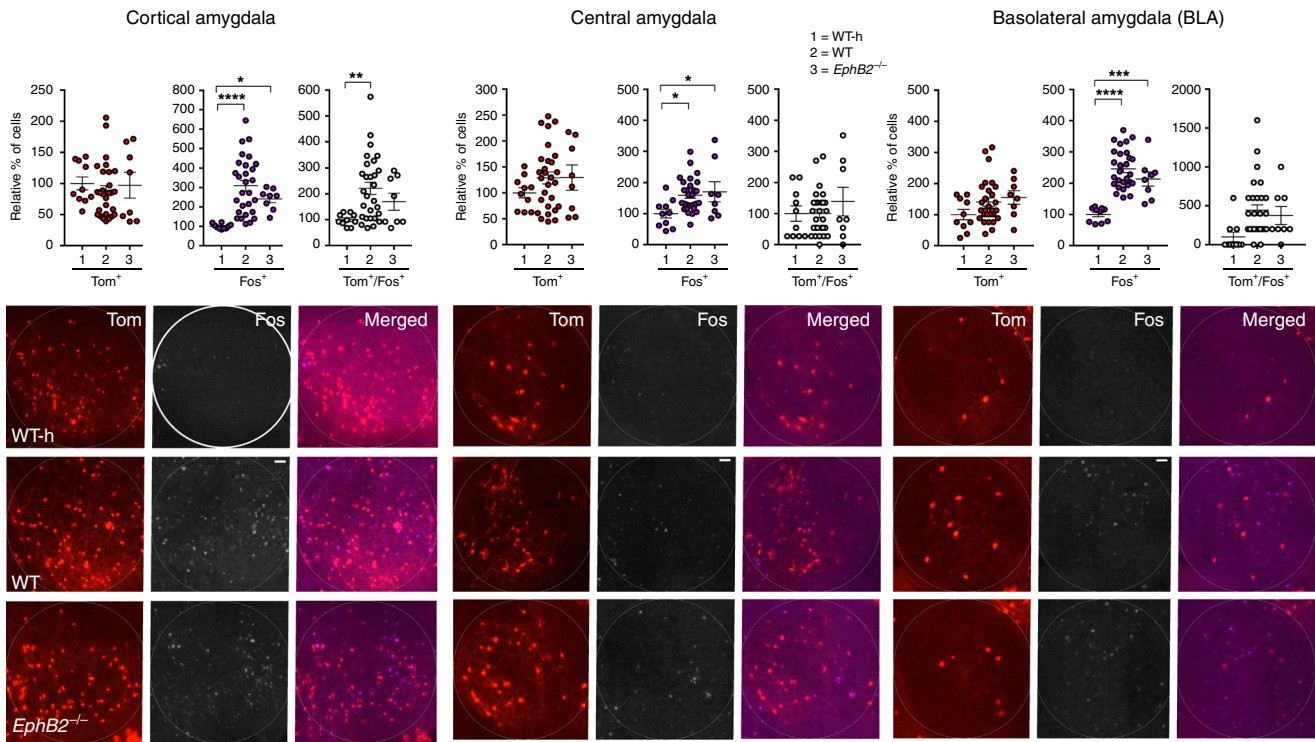

**Fig. 5** Analysis of Tom+ and Fos+ learning-associated neurons in the amygdala of *EphB2* mutant mice. Neurons activated exclusively following encoding (Tom+ single-positive), exclusively following sound-cued retrieval (Fos+ single-positive), and following both encoding and retrieval (Tom+/Fos+ double-positive) were imaged and counted in the cortical, central, and basolateral amygdala from *EphB2−/−* mutants and WT littermates subjected to FC and WT home cage controls not subjected to FC (WT-h). Scatter plots show the number of Tom+ single-positive neurons, Fos+ single-positive neurons, and Tom+/Fos+ double-positive neurons. Representative confocal images of Tom+ and Fos+ labeled neurons in the three amygdala regions are shown with the circle indicating quantification area. $n = 10$ hemisphere WT-h, $n = 28$ hemisphere WT, and $n = 8$ hemisphere *EphB2−/−* for all amygdala regions. For Fos+-single positive neurons in cortical amygdala, one-way ANOVA, $F_{(2,43)} = 11.65$, $p < 0.0001$; Tukey multiple comparison test; WT-h vs WT $q_{(36)} = 6.81$, $p < 0.0001$; WT-h vs *EphB2−/−* $q_{(16)} = 3.56$, $p = 0.0406$; WT vs *EphB2−/−* $q_{(34)} = 2.05$, $p = 0.3253$. For Tom+/Fos+-double positive neurons in cortical amygdala, one-way ANOVA, $F_{(2,43)} = 4.89$, $p = 0.0123$; Tukey multiple comparison test; WT-h vs WT $q_{(36)} = 4.37$, $p = 0.0096$; WT-h vs *EphB2−/−* $q_{(16)} = 1.95$, $p = 0.3619$; WT vs *EphB2−/−* $q_{(34)} = 1.71$, $p = 0.4539$. For Fos+-single positive neurons in central amygdala, one-way ANOVA, $F_{(2,43)} = 4.30$, $p = 0.0199$; Tukey multiple comparison test; WT-h vs WT $q_{(36)} = 3.83$, $p = 0.0259$; WT-h vs *EphB2−/−* $q_{(16)} = 3.50$, $p = 0.0448$; WT vs *EphB2−/−* $q_{(34)} = 0.63$, $p = 0.8975$. For Fos+-single positive neurons in basolateral amygdala, one-way ANOVA, $F_{(2,43)} = 24.76$, $p < 0.0001$; Tukey multiple comparison test; WT-h vs WT $q_{(36)} = 9.95$, $p < 0.0001$; WT-h vs *EphB2−/−* $q_{(16)} = 6.00$, $p = 0.0003$; WT vs *EphB2−/−* $q_{(34)} = 2.04$, $p = 0.3283$. Error bars are standard error of the mean (SEM); *$p < 0.05$, ***$p < 0.001$, ****$p < 0.0001$. Scale bar = 50 μm

**Table 1 Summary of changes in Tom+, Fos+, and Tom+/Fos+ neurons in WT mice subjected to fear conditioning compared to home caged WT mice (left columns) and in *EphB2−/−* mutant compared to WT both subjected to FC (right columns)**

| Cell types | WT compared to Home cage | | | *EphB2−/−* compared to WT | | |
|---|---|---|---|---|---|---|
| Brain regions | Tom+ | Fos+ | Tom+/Fos+ | Tom+ | Fos+ | Tom+/Fos+ |
| DG-upper blade | NC | ↑ | ND | NC | NC | ND |
| DG-lower blade | NC | NC | ND | NC | NC | ND |
| CA3 | NC | ↑ | ND | NC | NC | ND |
| CA1 | NC | ↑ | ND | NC | ↓ | ND |
| Auditory cortex | ↑ | ↑ | ↑ | NC | ↓ | NC |
| Cortical Amygdala | NC | ↑ | ↑ | NC | NC | NC |
| Central Amygdala | NC | ↑ | ND | NC | NC | ND |
| BLA | NC | ↑ | ND | NC | NC | ND |

*NC no change, ND none detected*

differences observed in neurons associated with learning and memory are actually due to the consequences of simple auditory stimulus exposure. Further, as we did not assess Tom/Fos staining in the auditory cortex of CS only control mice, it remains possible that tone-evoked responses may be altered in the *EphB2*-mutant mice and contribute to the observed changes in addition to those produced by fear conditioning. If this is the case, we would predict to observe reduced Fos levels in mutant auditory cortex after exposure to tone without fear conditioning, which cannot be disproved here as CS only mice would need to be assessed. Nevertheless, it is worth noting that EphB2 mutants have been shown to exhibit normal cochlear function[43]. It is also acknowledged that we have not shown the poor memory of *EphB2*-mutant mice is directly caused by specific dendritic/synaptic morphological abnormalities. It is therefore possible that the observed memory problems associated with these animals is due to an unidentified earlier developmental deficiency. Nevertheless, it is clear from our study that loss of EphB2 receptor function results in reduced context recall and tone recall and that the select group of neurons associated with such a learning/

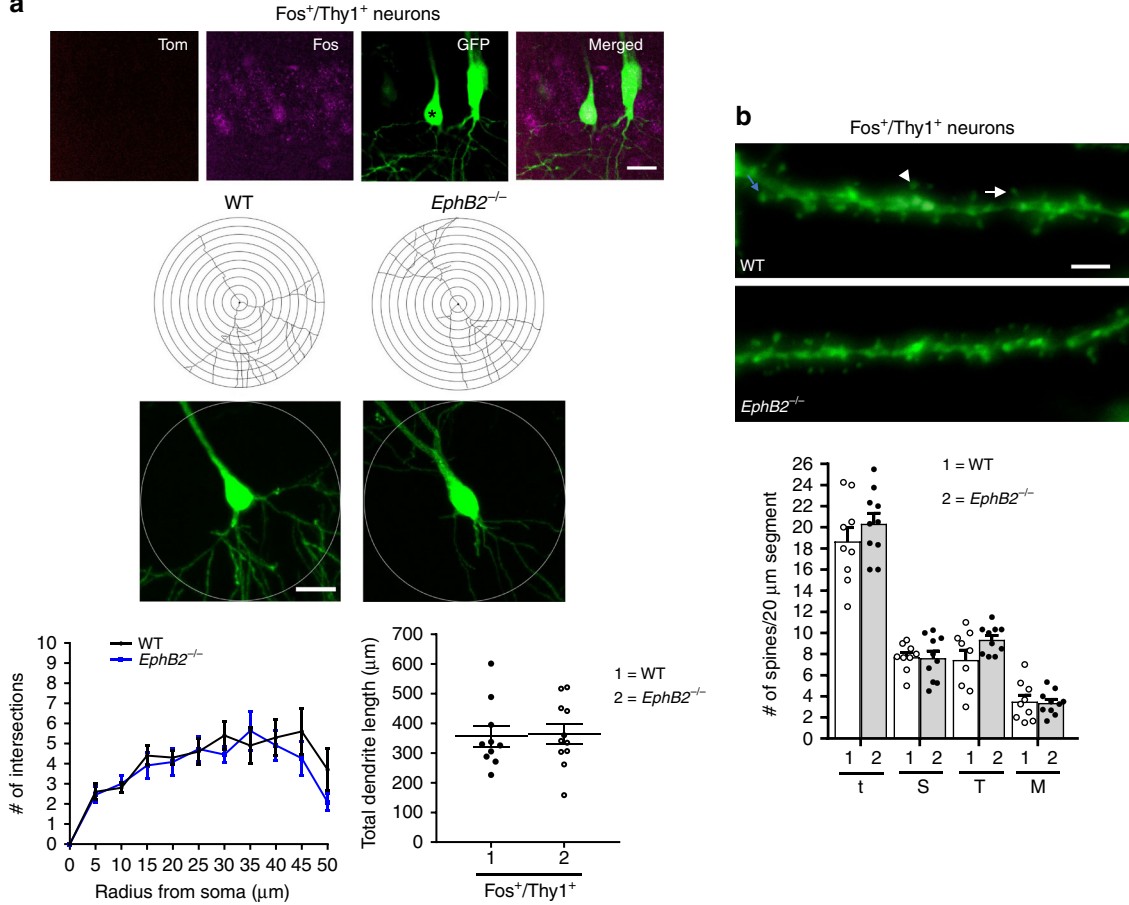

**Fig. 6** Analysis of dendritic branches and spine density/morphology in hippocampal CA1 of *EphB2* mutant mice. Confocal z-stacked images of Thy1-GFP[M] labeled neurons that were activated exclusively following the sound-cued retrieval test (Fos[+]/Thy1[+]) were obtained from the CA1 region. **a** Representative images of a Fos[+]/Thy1[+] neuron (asterisk), representative tracings of Thy1-GFP[M] labeled dendrites, and graphs of the quantified number of Sholl intersections and total dendritic length comparing WT and *EphB2[−/−]* mutant Fos[+]/Thy1[+] neurons. $n = 10$ neurons from three WT brains, and $n = 11$ neurons from three *EphB2[−/−]* brains; two-way repeated measures ANOVA. Circle diameter = 100 μm and scale bar = 20 μm. **b** Representative images of Fos[+]/Thy1[+] dendritic segments from WT and *EphB2[−/−]* mutant brains with arrows indicating mushroom shape (arrowhead) and thin (arrow) spines. Scatter plots of the quantified number of total spines (t) and breakdown into different morphologies of stubby (S), thin (T), and mushroom (M) shape spines. Spines from at least 3 × 20 μm dendrite segments per neuron were counted and averaged. $n = 9$ neurons from three WT brains, and $n = 10$ neurons from three *EphB2[−/−]* brains; two-way ANOVA. Scale bar = 2 μm

memory event (Tom[+]/Fos[+] neurons) fail to elaborate new and more mature spines.

The large family of Eph receptor tyrosine kinases and cognate membrane-anchored ephrin ligands transduce bidirectional signals into both the receptor-expressing cell (forward signaling) and ligand-expressing cell (reverse signaling)[40]. In the brain EphB and ephrin-B molecules are highly expressed in the hippocampus and cortex[9], and several studies have shown that trans-synaptic ephrin-B-EphB interactions promote synaptic development and function[23–27]. Our new data directly implicate EphB2 in the formation of long-term memories induced by the FC learning paradigm as loss of this protein leads to animals that perform poorly in both contextual and sound-evoked tests. The highly related and often co-expressed EphB1 receptor, however, is dispensable for such experience-mediated learning and memory. The function of EphB2 in learning and memory likely involves at least two avenues of forward signaling, that mediated by its tyrosine kinase catalytic activity and that mediated by its ability to bind PDZ domain proteins. As EphB2 binds directly to the NMDA receptor and induces its tyrosine phosphorylation[10,15], we anticipate that at least some of its actions in learning and memory involve

modulation of the NMDA receptor complex by aiding its postsynaptic localization/trafficking/stability, enhancing its calcium influx activity, inducing NMDA receptor-dependent IEG expression, and ultimately helping to activate neurons and drive LTP. Phosphorylation-dependent events likely play multiple important roles in regulation of the NMDA receptor and mediating learning/memory events, here EphB2 tyrosine kinase is utilized to enhance memories whereas serine phosphorylation mediated by cyclin-dependent kinase 5 (Cdk5) apparently acts as a negative factor by affecting NR2B cell surface levels[52]. Additionally, through its effects on the cytoskeleton, EphB2 has been implicated in dendritic branching and formation/remodeling/maturation of spines and synapses[16,17,19,20,22], which may involve EphB2 interactions with other synaptic partners including several PDZ domain-containing proteins (e.g., GRIP, Syntenin, PICK1), SH2/SH3 domain-containing proteins (e.g., Src, Abl, Grb2, RasGAP), and guanine nucleotide exchange factors (e.g., Vav, Tiam1, Ephexin, Intersectin, Kalirin).

At the cellular and molecular level, learning and memory is thought to be due to the orchestrated action of multiple protein players that modulate the connections between neurons and results in long-term structural changes to synapses. Our data

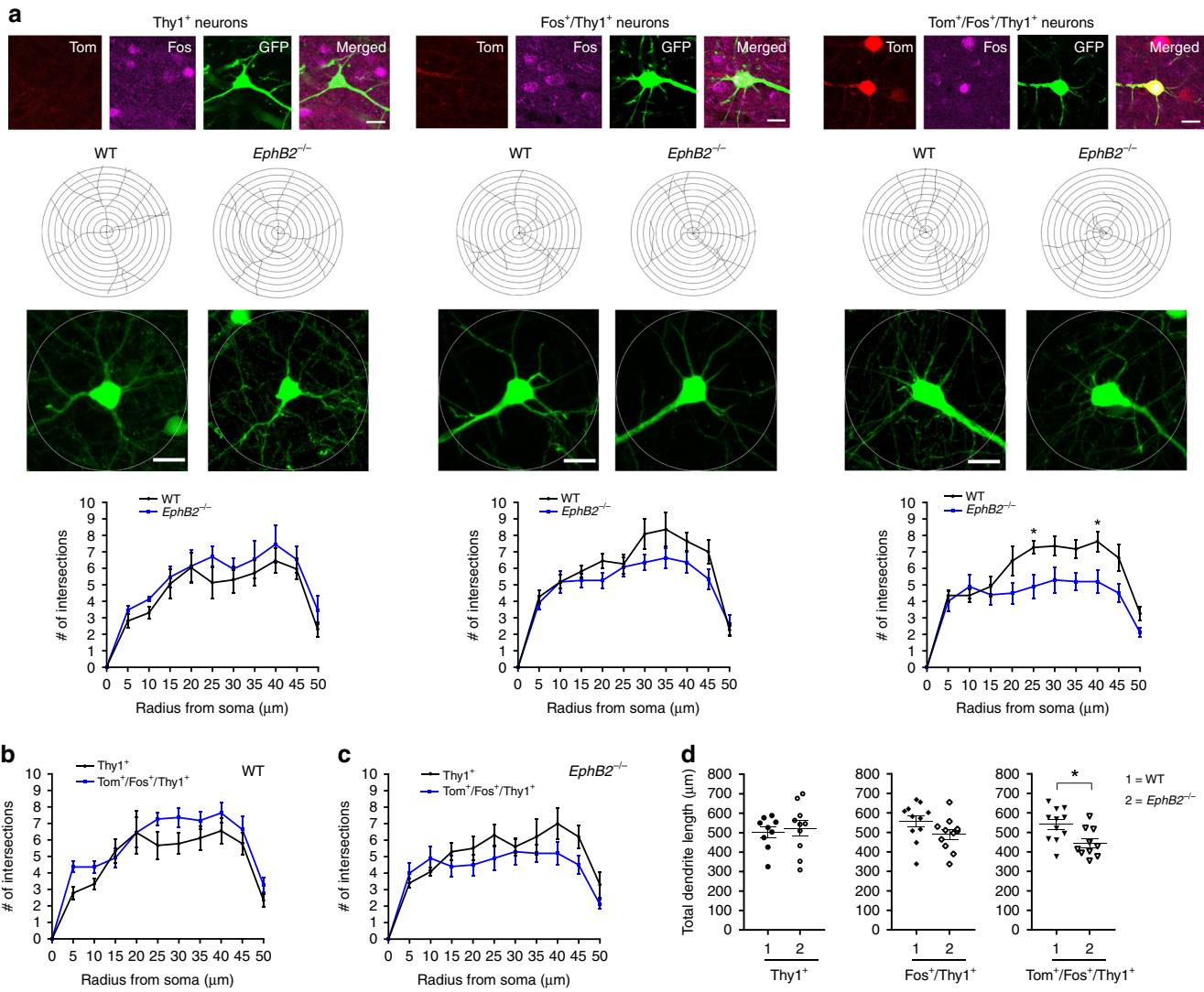

**Fig. 7** Analysis of dendritic branching in learning-associated neurons from the auditory cortex of *EphB2* mutant mice. Sholl analysis was performed on confocal z-stacked images of three different groups of Thy1-GFP^M labeled cortical neurons located in layers 3–4 of the auditory cortex, those that were Tom⁻/Fos⁻ dual-negative and not activated (Thy1⁺), those that were activated exclusively following the sound-cued retrieval test (Fos⁺/Thy1⁺), and those that were activated following both encoding and retrieval (Tom⁺/Fos⁺/Thy1⁺). Tom⁺ neurons only activated following encoding were rarely identified to colabel with Thy1⁺ and so were not analyzed. **a** Representative confocal images of the three different neuron groups assessed, representative tracings of Thy1-GFP^M labeled dendrites, and graphs of the quantified number of Sholl intersections between WT and *EphB2*^−/− mice. $n = 9$ neurons from three WT brains and $n = 10$ neurons from three *EphB2*^−/− brains (Thy1⁺ neurons), $n = 11$ neurons from three WT brains and $n = 11$ neurons from three *EphB2*^−/− brains (Fos⁺/Thy1⁺ neurons), and $n = 11$ neurons from three WT brains, and $n = 10$ neurons from three *EphB2*^−/− brains (Tom⁺/Fos⁺/Thy1⁺ neurons); two-way repeated measures ANOVA, interaction $F_{(10,190)} = 1.95$, $p = 0.0404$; effect of genotype $F_{(1,19)} = 11.56$, $p = 0.0030$; effect of repeated intersections $F_{(10,190)} = 26.19$, $p < 0.0001$; Bonferroni multiple comparisons tests; at radius 25 μm: $t_{(19)} = 2.9$, $p = 0.0457$; at radius 40 μm: $t_{(19)} = 2.98$, $p = 0.0359$. Circle diameter = 100 μm and scale bar = 20 μm. **b** Graph of the quantified number of Sholl intersections between Thy1⁺ and Tom⁺/Fos⁺/Thy1⁺ neurons in WT mice. Two-way repeated measures ANOVA. **c** Graph of the quantified number of Sholl intersections between Thy1⁺ and Tom⁺/Fos⁺/Thy1⁺ neurons in *EphB2*^−/− mice. Two-way repeated measures ANOVA. **d** Scatter plots of quantified total dendrite length of the three different neuron groups in WT and *EphB2*^−/− mice. Unpaired *t*-test; 540.5 μm ± 25.68 vs 443.8 μm ± 23.24; $t_{(19)} = 2.77$, $p = 0.0122$. Error bars are standard error of the mean (SEM); *$p < 0.05$

implicate EphB2 in mediating the structural plasticity of neurons associated with the learning of a new behavioral task. EphB2 therefore is a key trans-synaptic player necessary for the formation and long-term retention of learned information.

## Methods

**Mice**. The mutant and transgenic lines of mice used in this study have all been previously described: *EphB1*^−/− protein-null mutant[16], *EphB1*^lacZ/lacZ C-terminal intracellular truncated mutant[39], *EphB2*^−/− protein-null and *EphB2*^lacZ/lacZ C-terminal intracellular truncated mutations[40], *EphB2*^K661R/K661R kinase-dead

point mutation[41], *EphB2*^F620D/F620D kinase-overactive point mutation[42], and *EphB2*^ΔVEV/ΔVEV mutation that removes the three C-terminal amino acids to prevent EphB2 binding to PDZ domain containing proteins[41]. The *EphB*-mutant mice have been housed in the same animal facility together for years in a pigmented very robust mixed 129/CD1/C57Bl6 background to ensure all the strains/stocks thrive. Mutations are maintained by crossing *EphB*^+/mut heterozygous males and females to ensure litters have a mix of both *EphB*^mut/mut homozygotes and *EphB*^+/+ WT littermates for study. For double mutants, one gene is locked as a homozygote and the other as a heterozygote (e.g., *EphB1*^−/−;*EphB2*^+/−). The FosTrap mouse with CreER^T2 knocked into the *Fos* locus (*Fos*^Trap/+) generated by Liqun Luo[46,47] and Ai9 Cre indicator mouse with a silent tdTomato knocked into the *Rosa26* locus (*R26*^Ai9/+) generated by Hongkui Zeng[48] were both obtained

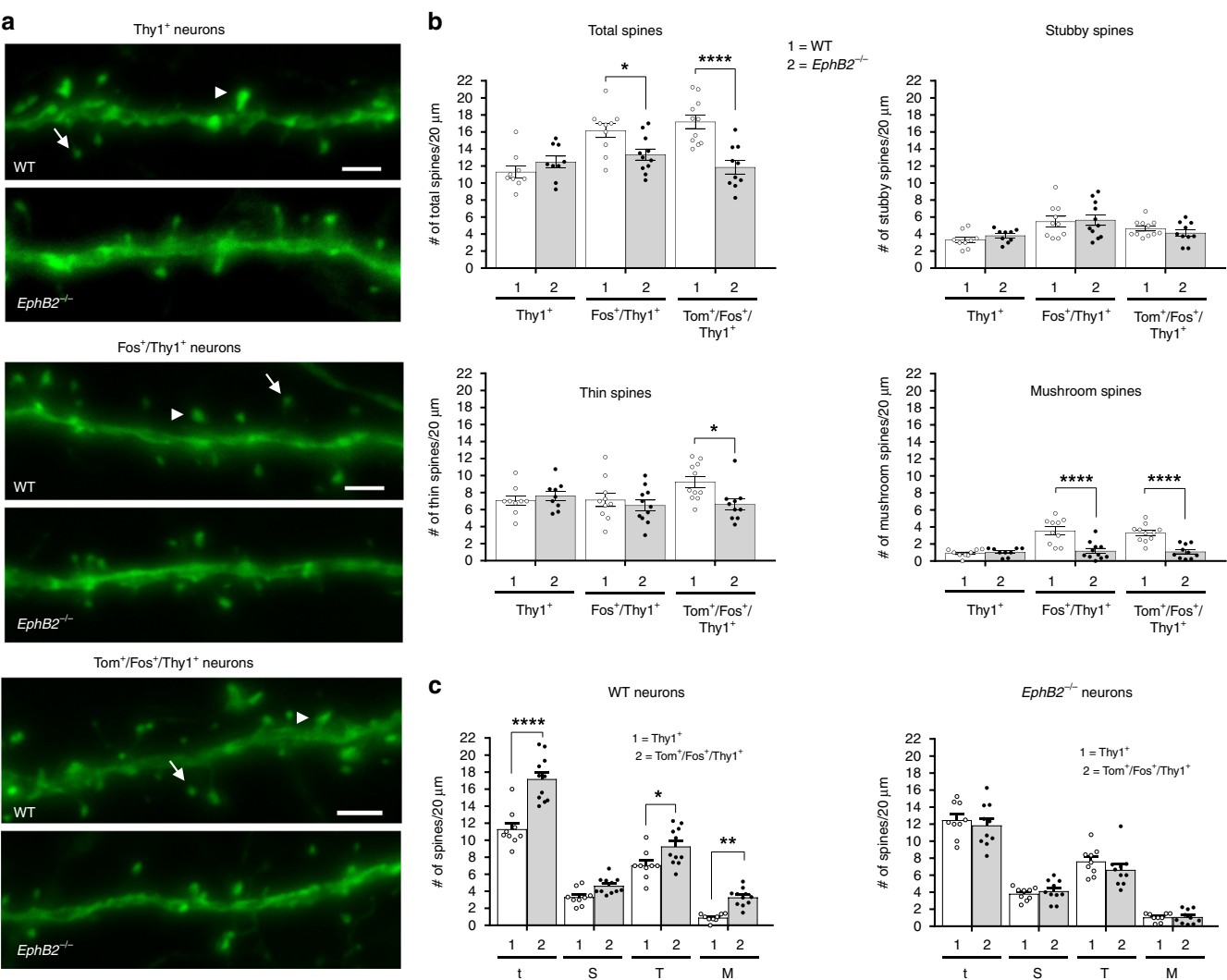

**Fig. 8** Analysis of spines in learning-associated neurons from the auditory cortex of *EphB2* mutant mice. Spines were analyzed from Thy1-GFP[M] labeled neurons in layers 3–4 of the auditory cortex that were either Tom−/Fos− dual-negative (Thy1+), Fos+ single-positive (Fos+/Thy1+), or Tom+/Fos+ dual-positive (Tom+/Fos+/Thy1+). **a** Representative confocal z-stack images of dendritic segments from WT and *EphB2−/−* brains identify mushroom (arrowhead) and thin (arrow) spines. Scale bar = 2 μm. **b** Scatter plots of the quantified number of total spines and breakdown into different morphologies of stubby, thin, and mushroom shaped spines from the three different groups of Thy1+, Fos+/Thy1+, and Tom+/Fos+/Thy1+ neurons, comparing WT and *EphB2−/−* mutants. Spines from at least 3 × 20 μm dendrite segments per neuron were counted and averaged. n = 9 neurons from three WT brains and n = 9 neurons from three *EphB2−/−* brains (Thy1+ neurons), n = 10 neurons from three WT brains and n = 11 neurons from three *EphB2−/−* brains (Fos+/ Thy1+ neurons), and n = 11 neurons from three WT brains, and n = 10 neurons from three *EphB2−/−* brains (Tom+/Fos+/Thy1+ neurons). For total spines, two-way ANOVA, interaction $F_{(2,54)} = 9.21$, $p = 0.0004$; effect of genotype $F_{(1,54)} = 14.43$, $p = 0.0004$; effect of neuron type $F_{(2,54)} = 8.52$, $p = 0.0006$; Sidak multiple comparison test; Fos+/GFP+ neurons $t_{(19)} = 2.73$; Tom+/Fos+/GFP+ neurons $t_{(19)} = 5.16$. For thin spines, two-way ANOVA, interaction $F_{(2,54)} = 2.99$, $p = 0.0588$; effect of genotype $F_{(1,54)} = 2.92$, $p = 0.0930$; effect of neuron type $F_{(2,54)} = 1.54$, $p = 0.2225$; Sidak multiple comparison test; Tom+/Fos+/GFP+ neurons $t_{(19)} = 2.93$. For mushroom shaped spines, two-way ANOVA, interaction $F_{(2,54)} = 9.91$, $p = 0.0002$; effect of genotype $F_{(1,54)} = 34.66$, $p < 0.0001$; effect of neuron type $F_{(2,54)} = 11.57$, $p = <0.0001$; Sidak multiple comparison test; Fos+/GFP+ neurons $t_{(19)} = 5.21$; Tom+/Fos+/ GFP+ neurons $t_{(19)} = 5.59$. **c** Scatter plots of the quantified number of total (t), stubby (S), thin (T), and mushroom (M) shaped spines, comparing Thy1+ neurons and Tom+/Fos+/Thy1+ neurons from WT (left) or *EphB2−/−* mutant (right) brains. For WT brain, two-way ANOVA, interaction $F_{(3,72)} = 7.15$, $p = 0.0003$; effect of neuron type $F_{(1,72)} = 60.95$, $p < 0.0001$; effect of spine shape $F_{(3,72)} = 204.10$, $p < 0.0001$; Bonferroni multiple comparison test; for total spines in WT $t_{(18)} = 7.8$; thin shaped spines in WT $t_{(18)} = 2.91$; mushroom shaped spines in WT $t_{(18)} = 3.16$. Error bars are standard error of the mean (SEM); *$p < 0.05$, **$p < 0.01$, ****$p < 0.0001$

from Jackson laboratory (Bar Harbor, ME, USA). The Ai9 element allows high-level ubiquitous expression of the red fluorescent protein tdTomato only after Cre-mediated the excision of the loxP-flanked transcriptional stop signal. The Thy1-GFP[M] reporter transgene mouse[49] was kindly provided by Josh Sanes. All experiments involving mice were carried out in accordance with the US National Institutes of Health Guide for the Care and Use of Animals under an Institutional Animal Care and Use Committee approved protocol and at an Association for Assessment and Accreditation of Laboratory Animal Care approved Facility at the University of Texas Southwestern Medical Center.

**Fear conditioning**. FC was done in the UT Southwestern Rodent Behavior Core facility by the director, Dr. Shari Birnbaum, who was blinded to the genotypes. A standard protocol was followed to ensure all mice were exposed to the same pairing of a conditioned stimulus (CS, context and sound) with an unconditioned stimulus (US, foot shock):

*Day 0 (encoding/training)*: Early in the morning group housed unisex cages (both males and females were tested at age 3–5 months) were quietly wheeled while covered from Henkemeyer colony a short distance to Behavior Core and placed for at least 3 h in a small dimly lit closet immediately adjacent to FC procedure room

**Table 2 Summary of raw numbers of Tom$^+$, Fos$^+$, and Tom$^+$/Fos$^+$ neurons in home caged WT mice, and in WT and EphB2$^{-/-}$ mutant mice subjected to fear conditioning**

| Cell types | Home-cage WT (mean ± SEM) | | | WT (mean ± SEM) | | | EphB2$^{-/-}$ (mean ± SEM) | | |
| Brain regions | Tom$^+$ | Fos$^+$ | Tom$^+$/Fos$^+$ | Tom$^+$ | Fos$^+$ | Tom$^+$/Fos$^+$ | Tom$^+$ | Fos$^+$ | Tom$^+$/Fos$^+$ |
|---|---|---|---|---|---|---|---|---|---|
| DG-upper blade | 4.4 ± 0.43 | 9.5 ± 0.43 | 0.07 ± 0.04 | 5.76 ± 0.43 | 16.34 ± 0.6 | 0.09 ± 0.04 | 6.08 ± 0.58 | 15.25 ± 2.07 | 0.042 ± 0.04 |
| DG-lower blade | 2.63 ± 0.35 | 7.73 ± 0.68 | 0 ± 0 | 2.92 ± 0.27 | 9.49 ± 0.59 | 0 ± 0 | 2.46 ± 0.45 | 7.21 ± 1.21 | 0 ± 0 |
| CA3 | 1.07 ± 0.36 | 20.1 ± 0.98 | 0.23 ± 0.11 | 0.78 ± 0.13 | 27.2 ± 1.03 | 0.09 ± 0.03 | 1.25 ± 0.27 | 25.04 ± 4.15 | 0.08 ± 0.08 |
| CA1 | 0.98 ± 0.16 | 7.26 ± 0.54 | 0.07 ± 0.03 | 1.17 ± 0.12 | 17.19 ± 0.8 | 0.09 ± 0.02 | 1.28 ± 0.17 | 12.67 ± 0.8 | 0.17 ± 0.04 |
| Auditory cortex | 47.61 ± 3.21 | 18.76 ± 1.97 | 3.33 ± 0.64 | 62.2 ± 2.01 | 105.15 ± 7.74 | 16.13 ± 1.18 | 62.4 ± 2.37 | 74 ± 9.65 | 13.22 ± 1.42 |
| Cortical Amygdala | 52.63 ± 5.47 | 21.97 ± 0.89 | 5.4 ± 0.39 | 46.62 ± 4.31 | 68.09 ± 6.08 | 11.96 ± 1.28 | 51.12 ± 10.91 | 53 ± 4.25 | 9.17 ± 1.79 |
| Central Amygdala | 26.23 ± 2.61 | 11.4 ± 1.61 | 1.23 ± 0.3 | 33.79 ± 3.03 | 18.20 ± 1.15 | 1.24 ± 0.16 | 33.96 ± 6.4 | 19.42 ± 3.7 | 1.71 ± 0.57 |
| BLA | 2.63 ± 0.43 | 17.5 ± 1.3 | 0.17 ± 0.1 | 3.7 ± 0.35 | 43.14 ± 2.04 | 0.74 ± 0.12 | 4.08 ± 0.58 | 37.42 ± 4.02 | 0.62 ± 0.19 |

to minimize stimulation. Individual mice were then placed in a FC chamber A, which is a white box with a metal bar floor to provide electric shock, a speaker, and a video camera connected to software that measures movement/freezing. For encoding step, mice were allowed to explore chamber for 120′′ and was then subjected to three cycles of 30′′ white noise tone 80 dB co-terminated with a 2′′ shock (0.5 mA) and 60′′ rest between cycles 1 and 2, ending with a 30′′ rest period after final shock (6 min total in chamber). Mice were then returned to their home cage and placed back in dimly lit closet. The chamber, metal bars, and waste tray was disinfected and dried between runs. In the late afternoon, at least 3 h after last mice were trained, cages were covered and gently wheeled back to Henkemeyer colony. A separate set of WT mice were used in a control CS only experiment to ensure general handling and exposure to the training context and sound stimuli (in the absence of shocks) does not lead to a elevated freezing response compared to CS-US trained mice. In the FC experiments that included $Fos^{Trap}$ and $R26^{Ai9}$ elements, a single intraperitoneal (IP) injection of 4-hydroxytamoxifen (4-OHT, 50 mg/Kg in corn oil, 10 mg/ml) was given to the mouse immediately prior to it being placed in the FC chamber and subjected to encoding step. For the Trapping experiments, control home cage mice that received 4-OHT injection but not subjected to the FC protocol were also included in the analysis.

*Day 2 (context-cue retrieval/recall test)*: Early in morning cages were covered and gently transported to the dimly lit closet immediately adjacent to FC procedure room to minimize stimulation for at least 3 h. Mice were then individually placed in FC chamber A and freezing time monitored for 5 min, after which they were returned to their cage and placed back in closet until late afternoon when all cages were covered and wheeled back to Henkemeyer colony.

*Day 4 (sound-cue retrieval/recall test)*: Early in morning mice were covered and gently transported to the dimly lit closet immediately adjacent to FC procedure room to minimize stimulation for at least 3 h. Mice were then placed in a modified FC chamber B (white board covering shock bars, black triangle roof, vanilla scent) and freezing time monitored for 3 min with no tone (pre-tone) followed by 3 min with white noise tone (post-tone). Mice were then returned to their home cage and placed back in dimly lit closet. For the FC experiments that included FosTrap/Ai9/Thy1-GFP$^M$, exactly 90 min after being returned to their home cage to allow for IEG expression, mice were anesthetized using a 9:1:10 ketamine/xylazine/PBS mix that delivers 225 mg/kg ketamine (45 mg/ml)/25 mg/kg xylazine (5 mg/ml), and brains were fixed by cardiac perfusion with PBS followed by 4% paraformaldehyde in PBS. Brains were then dissected, post-fixed overnight in 4% paraformaldehyde in the dark at 4 °C, washed three times with PBS, and stored in PBS containing 0.05% sodium azide in the dark at 4 °C. Following verification of genotypes, brains were provided to A.T. for blinded analysis.

**Brain preparation and immunofluorescence.** Fixed brains were embedded in 3% agarose and 50 μm coronal sections cut with a vibratome (frequency 7 Hz, speed 5 Hz). Slices were selected between interaural 2.36–1.64 mm (Bregma −1.43 to −2.15 mm) with ~700 μm thickness giving 12–15 slices. Slices were placed in 24-well plates (1–2 slices in each) in PBS containing 0.05% sodium azide and maintained in the dark at 4 °C.

Free floating vibratome sections were placed in 24-well plates in blocking solution (4% donkey serum, 4% goat serum, 0.1% Triton X-100 in PBS) for 1 h at RT and then probed with 1:500 dilution of rabbit anti-c-Fos 9F6 monoclonal antibody (Cell Signaling, #2250) in blocking solution overnight at 4 °C. Sections were then washed 3 × 10 min in PBST (PBS containing 1% Tween-20) and probed with 1:500 dilution of donkey anti-rabbit Alexa fluor 647-conjugated secondary antibody (Jackson ImmunoResearch, #711-605-152) and a 1:250 dilution of 0.02 mg/ml DAPI (Sigma, #D9542) in blocking solution for 1 h at RT. Sections were then washed in PBST for 3 × 10 min and mounted on charged slides using an aqueous mounting solution (Thermo Scientific, Immu-Mount #9990402).

**Neuron counts and analysis of dendrites and spines.** Neurons were imaged using a Zeiss LSM710 confocal laser-scanning microscope and quantified from three sections of each hemisphere in defined areas of the hippocampus, auditory

cortex, and amygdala. The size of each area counted are as follow: 0.32 mm$^2$ rectangle in CA1, 0.32 mm$^2$ rectangle in DG upper blade and lower blade, 0.42 mm$^2$ rectangle in CA3, 0.42 mm$^2$ rectangle in auditory cortex, and 0.33 mm$^2$ circle in BLA, central amygdala, and cortical amygdala.

For cell population assessment, the number of tdTomato fluorescent-labeled neurons (Tom$^+$) indicative of FosTrap activity following encoding stage, the number of Fos antibody-labeled newly activated neurons (Fos$^+$) following retrieval stage, and the number of double-labeled Tom$^+$/Fos$^+$ neurons were counted in given areas of hippocampus, auditory cortex and amygdala. As the anti-Fos antibody gives a range of signals from a few small punctate particles to fully filled nuclear staining, any cell that exhibited less than 5 particles was considered a Fos-negative neuron.

Dendrite branching (Sholl analysis) and spine density/morphology was assessed in Thy1-GFP$^M$ fluorescent-labeled neurons. High resolution z-stacks were obtained from selected Tom$^+$/Fos$^+$/Thy1$^+$ triple-labeled, Fos$^+$/Thy1$^+$ double-labeled, and Thy1$^+$ single-labeled neurons using ×63 objective, merging 8–10 images at 1.5 μm interval (for Sholl analysis) and 0.5 μm interval (for spine analysis). At least three neurons of each class were analyzed per brain in the areas of interest. Sholl intersections were counted at ten 5 μm rings that extended to a 50 μm radius drawn around selected soma.

Spine density and morphology was assessed from 20 μm segments of at least one apical and one basal branch of an identified neuron (at least three segments per neuron). Spine classification was defined with simplified head/neck ratio = 2 and stem/neck ratio = 2 as follows; mushroom spines: spines with a head diameter more than two times of neck diameter and a stem length less than two times of neck diameter, thin spines: spines with a head diameter less than 2 times of neck diameter and a stem length more than two times of neck diameter, stubby spines: spines with no head, and a stem length less than two times of neck diameter.

**Statistics and reproducibility.** Neuron counts and spine analysis was performed using ImageJ and Sholl analysis was performed using Fiji (Rasband, National Institutes of Health). For neuron counts in each region of the brain assessed, the average number of Tom$^+$ single-positive cells, Fos$^+$ single-positive cells, and Tom$^+$/Fos$^+$ double-positive cells that were detected in the WT home cage control mice was set to 100% and counts for corresponding WT and EphB2$^{-/-}$-mutant fear conditioned mice made relative to that. A summary of the differences in cell counts between the WT home cage controls, and the fear conditioned WT and EphB2$^{-/-}$-mutant mice is shown in Table 1 and the actual means ± s.e.m. (standard error of the mean) for all cell counts are provided in Table 2. Statistical analysis was done using GraphPad Prism 7 and either two-tailed paired or unpaired student's t-tests, one-way ANOVA, two-way ANOVA, or repeated measures ANOVA with post-hoc analysis using either Tukey, Dunnett, Bonferroni, or Sidak tests. The exact number of mouse brains or neurons assessed, the statistical test used, t value or F value, degrees of freedom, and exact p-values are provided in each figure legend. A p-value < 0.05 was considered a significant difference between means, with the range of p values in each comparison shown by asterisks in the graphs (*p < 0.05, **p < 0.01, ***p < 0.001, ****p < 0.0001).

**Reporting summary.** Further information on research design is available in the Nature Research Reporting Summary linked to this article.

## Data availability
The authors declare that all data supporting the findings of this study are available within the article and in the supplementary figures. Source data for Figs. 1–8 can be found in Supplementary Data 1.

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

## Acknowledgements

We thank Dr. Shari Birnbaum for help with the fear conditioning experiments and Francis Sprouse and Rachel Britton for help with genotyping and cardiac perfusions. This research was supported by the BrightFocus Foundation and the NIH (MH066332) to M.H.

## Author contributions

All the animal work including husbandry and fear conditioning was done by M.H., all the brain analysis was done by A.T., and both authors analyzed the data and wrote the manuscript.

## Competing interests

The authors declare no competing interests.
