## [Peer Review File · Communications Biology]

Reviewers' comments:

Reviewer #1 (Remarks to the Author):

The goal of the study is to understand the role of EphB receptor forward signaling in FC learning and memory recall. These studies use several different EphB1 and EphB2 mutant mice to elucidate the role of EphB forward signaling and suggest possible role of EphB2 receptor in retrieval of fear-conditioned memory. The findings are interesting and suggest impaired contextual and tone recall in EphB2^{-/-} mice, most likely due to forward signaling through the receptor intracellular domain. Observed changes in the number of cFos positive cells in the auditory cortex following tone recall are also intriguing, but it is not clear whether the changes are due to different responses to tone or tone memory retrieval. Therefore, potential impact of the presented findings is limited due to lack of appropriate CS only control for EphB2^{-/-} mice in order to interpret the results of both behaviour tests and cFos⁺ cell analysis. Author use an interesting approach to label activated cFos⁺ cells during training and then again during memory recall. However, use of tamoxifen during learning may complicate the interpretations of the results. Additional experiments and analysis using tamoxifen-treated animals without FC should be performed to support their statements. I believe that the manuscript needs major revisions before it is acceptable for publication.

Major concerns:

1. CS only control is missing for EphB2^{-/-} mice. While there is no difference is observed between CS-US WT and CS-US KO mice using post-hoc analysis ($p=0.06$), the effect of genotype is significant with 2-way ANOVA (fig1B, $p=0.0198$). Do CS-US KO freeze more compared to CS KO mice?
2. In fig2, appropriate controls, CS WT and CS EphB2^{-/-} mice, are missing. It would be more appropriate to use CS only mice instead of home cage controls here as we know exposure of mice to context can trigger activation of cells in the hippocampus even without FC. It is inappropriate to compare FC KO to WT home cage group. KO home cage group is missing here as well. Four groups should be analyzed to examine the effects of FC and genotype using 2-way ANOVA.
3. Tone-evoked cell activation in the auditory cortex may be different between WT and B2^{-/-} mice and contribute to the changes in cFos⁺ cell density that are reported here. As no differences are seen in the density of Tom⁺ cells, but only cFos⁺ cells stained after tone presentation.
4. Acute IP injection of tamoxifen is shown to affect learning and memory through the activation of ER, therefore it is important to have tamoxifen-injected control without FC. If females used in this study, female estrous cycle should be controlled for, especially because tamoxifen was used.
5. In fig7 Important control is missing. Were dendritic branches are different between CS only EphB2^{-/-} and WT auditory cortex? Exposure to tone can also affect spines, especially if there are baseline differences in spine density or the number of activated cells in the auditory cortex of B2^{-/-} mice.
6. In discussion, it is not clear if decreased complexity is a result of differences induced by tone retrieval following FC or there are also baseline differences in the absence of training.

Minor concerns:

1. Use of long-term memory (LTM) would not be appropriate here as memory retrieval is tested 2 days after the training. Instead it should be called contextual recall or tone recall.

2. In fig1B, graphs should show freezing during all three training sessions.
3. cFos staining is hard to see in blue channel (fig2). Green channel should be used for better visualization of cFos and co-localization
4. In fig3, images are poor quality. It is hard to see both red and blue cells.
5. In fig4, were the layer specific differences observed?
6. In fig4, Tom+ image for EphB2^{-/-} shows low number of red cells, which contradicts results presented in graphs. cFos cells are also hard to see in blue channel.

Reviewer #2 (Remarks to the Author):

EphrinB receptors and their cognate ligand partners form neuronal signalling complexes linked via tyrosine kinases that act in forward as well retrograde directions. These complexes have been found to have distinct roles in development as well as synaptic function at mature synapses. In this study, the authors study EphB2 receptor signalling, and its role in fear learning, using a range of different transgenic animals. They use a classical fear learning paradigm, contextual and cued (sound) fear to study learning and memory in mice in which the EphB2 receptors have been genetically deleted, or their forward signalling has been disrupted. They show firstly that in mice in which EphB2 tyrosine kinase signalling has been disrupted, fear conditioning to both context and tone is greatly reduced. Using anatomical analysis, they show that in animals where EphB2 signalling is disrupted there are differences in the anatomical changes in identified neurons.

The major conclusion of this study is: "We show here that EphB2 receptor forward signaling is necessary for FC-induced learning and memory, with both its intracellular tyrosine kinase catalytic activity and ability to couple to PDZ domain containing proteins being particularly important for sound-cued, hippocampal-independent memories".

Overall, the experiments are well done, and the data carefully addressed. It is clear that in when EphB2 signalling is disrupted, animals have a deficit in fear learning. This effect is replicated in animals where forward tyrosine kinase signalling is specifically disrupted. These behavioural changes are mimicked by morphological changes that are seen following fear learning. Moreover, in a strain where forward tyrosine kinase signalling is enhanced there may be an enhancement in learning.

This is a very extensive data set. However, I cannot see how these data lead to the conclusions that are reached. This study clearly shows that when EphB2 signalling is disrupted, animals do not learn well. Moreover, there are clearly morphological changes that are identified in neurons that are likely engaged during learning and/or retrieval using an immediate early gene marker. However, the link between these two sets of observations is not shown to be causal.

All experiments were done comparing WT mice with those that harbour one or several mutated genes. Thus, it seems equally likely that EphB2 signalling is not in fact engaged during learning or retrieval, but, as these are transgenic strains, the lack of EphB2 signalling leads to developmental changes that block fear learning and molecular events that follow it. With regard to the Fos⁺ labelling, they themselves suggest that they cannot rule out the possibility that this is simply labelling sensory

activated neurons. The impact of disrupting EphB2 function on synaptic or neural function are never explored. For instance, they assume that the morphological changes they describe are due to changes in synaptic plasticity - however, this is not explored for the cells in question.

Other points:

For the neuron counts in the method it is stated that

"For neuron counts in each region of the brain assessed, the average number of Tom+ single-positive cells, Fos+ single positive cells, and Tom+/Fos+ double-positive cells that were detected in the WT home cage control mice was set to 100%".

I found this somewhat confusing. When looking at the figures (eg Fig 2, upper blade), in the first panel counts are presented for TOM+ cells for wild-type in the home cage (WT-h - labelled #1), wild type that underwent fear conditioning (WT - #2), and EphB2-/- (#3). For the WT-h animals I expected this to be a 100% - however, as can be seen the points are scattered around the 100%. And similarly for the Fos+ single positive and the Tom+/Fos+ counts.

Reviewers' comments:

Reviewer #1 (Remarks to the Author):

The goal of the study is to understand the role of EphB receptor forward signaling in FC learning and memory recall. These studies use several different EphB1 and EphB2 mutant mice to elucidate the role of EphB forward signaling and suggest possible role of EphB2 receptor in retrieval of fear-conditioned memory. The findings are interesting and suggest impaired contextual and tone recall in EphB2^{-/-} mice, most likely due to forward signaling through the receptor intracellular domain. Observed changes in the number of cFos positive cells in the auditory cortex following tone recall are also intriguing, but it is not clear whether the changes are due to different responses to tone or tone memory retrieval. Therefore, potential impact of the presented findings is limited due to lack of appropriate CS only control for EphB2^{-/-} mice in order to interpret the results of both behaviour tests and cFos⁺ cell analysis. Author use an interesting approach to label activated cFos⁺ cells during training and then again during memory recall. However, use of tamoxifen during learning may complicate the interpretations of the results. Additional experiments and analysis using tamoxifen-treated animals without FC should be performed to support their statements. I believe that the manuscript needs major revisions before it is acceptable for publication.

Major concerns:

1. CS only control is missing for EphB2^{-/-} mice. While there is no difference is observed between CS-US WT and CS-US EphB2 mutant mice using post-hoc analysis ($p=0.06$), the effect of genotype is significant with 2-way ANOVA (fig1B, $p=0.0198$). Do CS-US EphB2 mutant freeze more compared to CS EphB2 mutant mice?

The data shown in Figure 1b was obtained at the time of FC training on day 0 and is used to simply show the EphB2 mutant mice are responsive to the encoding/training exercise. Specifically, learning during FC training was determined by measuring the percentage of time each mouse froze during the initial 120" period prior to the first sound/shock cycle (pre-tone) and comparing to the percentage of time they froze during the final 30" period following the third foot shock (post-tone). The data indicate that EphB2 mutant mice, like their WT counterparts, initially learn during the FC training by exhibiting a significant increase in percentage of time freezing following the CS-US training protocol compared to before they were subjected to the sound/shock cycles. The data further indicates that there are no significant differences between pre-tone WT and mutant, nor are there difference between post-tone WT and mutant. And while the two-way ANOVA may indicate effect of genotype is significant as reviewer points out, the necessary post-hoc analysis indicate that there are no significant effects of genotype. In contrast, both two-way ANOVA and post-hoc analysis indicate the differences between pre-tone and post-

tone are highly significant for both the WT and the EphB2 mutant animals. Additionally, please note that in this experiment the pre-tone is actually like a CS only control, and the post-tone is level of freezing in paired CS-US. So, yes, the EphB2 mutant paired CS-US freezes significantly more than EphB2 mutant CS only. Related to this issue, in response to another question from this reviewer (below), we also provide in the revised manuscript the freezing data following each of three sound/shock cycles during the training session as Supplementary Figure 1. This additional data is consistent with the idea that the WT and EphB2 mutants increase their level of freezing during the training session.

2. In fig2, appropriate controls, CS WT and CS EphB2^{-/-} mice, are missing. It would be more appropriate to use CS only mice instead of home cage controls here as we know exposure of mice to context can trigger activation of cells in the hippocampus even without FC. It is inappropriate to compare FC EphB2 mutant to WT home cage group. EphB2 mutant home cage group is missing here as well. Four groups should be analyzed to examine the effects of FC and genotype using 2-way ANOVA.

Reviewer asks for us to repeat all experiments to include CS only controls instead of using WT home cage controls. Reviewer seems to miss the point. The important data in Figure 2-5 are not comparing home cage controls to WT CS-US, but rather comparing WT CS-US mice to EphB2 mutant CS-US mice. This is the informative data which we focus our study. The home cage controls are simply to show that there are clear/robust increases in Tom⁺ and Fos⁺ neurons in certain regions of the brain in the WT mice subjected to CS-US. What is important in our analysis and what we focus attention on are regions where the WT CS-US neuron counts are significantly different from EphB2 mutant CS-US, which we show are observed only in Fos⁺-single positive neurons in the CA1 and the auditory cortex. The only reason we included home cage data is to demonstrate that Fos⁺-positive cells (DG, CA1, CA3, amygdala, auditory cortex), Tom⁺-positive cells (auditory cortex only) and Fos⁺/Tom⁺ double-positive cells (cortical amygdala, auditory cortex) are increased as expected in the mice exposed to the CS-US protocol. We believe that spending time running multiple control experiments will not in any way change the outcome of our study, which is focused on comparing WT CS-US mice to EphB2 mutant CS-US mice.

3. Tone-evoked cell activation in the auditory cortex may be different between WT and B2^{-/-} mice and contribute to the changes in cFos⁺ cell density that are reported here. As no differences are seen in the density of Tom⁺ cells, but only cFos⁺ cells stained after tone presentation.

Reviewer is not pointing out any concern here. In fact, reviewer agrees with our line of thinking that the key informative data is comparing WT CS-US mice to EphB2 mutant CS-US mice.

4. Acute IP injection of tamoxifen is shown to affect learning and memory through the activation of ER, therefore it is important to have tamoxifen-injected control without FC. If females used in this study, female estrous cycle should be controlled

for, especially because tamoxifen was used.

Reviewer raised concerns about tamoxifen, which is debatable depending on which papers you read as some indicate an effect and others indicate no effect. Nevertheless, all animals in the manuscript used for data shown in Figures 2-8 each received a single injection of tamoxifen. Thus, tamoxifen is not a variable in this data. Nevertheless, we have isolated the tamoxifen injected WT and EphB2 mutant freezing data from mice that did not receive tamoxifen. No significant differences were observed. The graphs are provided as Supplementary Figure 3 in the revised version of the manuscript. We have also separated male from female freezing data and did not observe any sex differences except for F620D mutant mice in contextual FC and dVEV mutant in sound-cued, and include this information as Supplementary Figure 2 in the revised version of the manuscript. While further study of such sex differences may be worthy to follow up in the future, such additional analysis would be beyond the scope of the current study we are trying to get published and will not in any way change the main take home message of our study.

5. In fig7 Important control is missing. Were dendritic branches are different between CS only EphB2^{-/-} and WT auditory cortex? Exposure to tone can also affect spines, especially if there are baseline differences in spine density or the number of activated cells in the auditory cortex of B2^{-/-} mice.

All mice involved in Figures 6-8 are subjected to CS-US, thus CS is not a variable in the analysis. Here, rather, it is important to focus on comparing WT CS-US to EphB2 mutant CS-US in the three different and distinct neuron classes we assessed (Tom⁺/Fos⁺, Tom⁻/Fos⁺ and Tom⁻/Fos⁻). In fact, the Tom⁻/Fos⁻ neurons are very much a perfect internal control that centers on neurons that were neither activated during training nor during recall. If there were “baseline” differences between WT and EphB2 mutants, then those would have definitely been revealed in our scoring for Thy1-GFP-M labeled dendrites/spines in these Tom⁻/Fos⁻ neurons. Importantly, however, we observed no differences in dendritic branches or spines in Tom⁻/Fos⁻ neurons between WT and EphB2 mutants. However, as shown in Fig. 8b and 8c, striking and highly significant increases in spine density and maturation were observed in Tom⁺/Fos⁺ and Tom⁻/Fos⁺ neurons, but only in the WT brains. The corresponding Tom⁺/Fos⁺ and Tom⁻/Fos⁺ neurons in the EphB2 mutant brains did not exhibit any increases in complexity whatsoever and instead looked no different from the Tom⁻/Fos⁻ untrained “baseline” neurons. With this data, we hypothesize the presence of EphB2 receptor protein (in the WT brain) is necessary for the elaboration of new additional spines as well as increases in the number of mature mushroom and thin spines in the select group of Tom⁺/Fos⁺ and Tom⁻/Fos⁺ learning-associated neurons. Because the study already incorporates a much better internal negative control in the Tom⁻/Fos⁻ neurons, we believe the additional negative control experiments that reviewer requests are unnecessary and will not alter the main conclusions in any meaningful way.

6. In discussion, it is not clear if decreased complexity is a result of differences induced by tone retrieval following FC or there are also baseline differences in the absence of training.

Concern is whether decreased complexity is due to fear conditioning or if EphB2 mutant mice show decreases in the absence of training. As described in response to point #5 above, the Tom⁻/Fos⁻ neurons labeled with Thy1-GFP-M in WT and EphB2 mutant mice show absolutely no differences in dendritic branches or spines. These Tom⁻/Fos⁻ neurons are the perfect baseline internal controls that show the complexity of untrained neurons not involved in the fear conditioning is no different between the WT and EphB2 mutant brains.

Minor concerns:

1. Use of long-term memory (LTM) would not be appropriate here as memory retrieval is tested 2 days after the training. Instead it should be called contextual recall or tone recall.

LTM is widely used in the literature to describe the overall ability of an animal to remember what was learned for an extended period of time, typically after a few hours and lasting days or longer. We tested contextual memory after 2 days and sound-cued memory after 4 days, and thus we considered these periods of remembering as long-term. Recall is perhaps more specific for a testing event while LTM is perhaps a general term to define the ability to learn and remember for a long period of time. Nevertheless, in our manuscript, we already used both terms and used LTM for an overall effect and used recall when discussing specific event or test.

2. In fig1B, graphs should show freezing during all three training sessions.

As mentioned above, we provide in the revised manuscript a more detailed graph for freezing data obtained during the encoding/training session on day 0 of our study (please see the new Supplementary Figure S1).

3. cFos staining is hard to see in blue channel (fig2). Green channel should be used for better visualization of cFos and co-localization

Reviewer wants us to repeat c-Fos staining using green channel (we used Alexa fluor 647-conjugated secondary antibody which gives purple fluorescence). As these brains also contained the Thy1-GFP-M green fluorescent reporter to visualize dendrites and spines, it was not possible to visualize the c-Fos antibody staining with a green secondary, which probably would have given us stronger signals. Importantly, however, we did provide high magnification images in Figure 4 that show individual purple Fos⁺ neurons, individual Tom⁺ neurons, and double-labeled Fos⁺/Tom⁺ neurons in the auditory cortex. We could artificially change the color but would rather not manipulate the data. If the reviewer simply increases the magnification on PDF viewer, the Fos⁺ cells are easy to see.

4. In fig3, images are poor quality. It is hard to see both red and blue cells.

Reviewer says that it is hard to see purple cells in Figure 3. Here, in the CA1 and CA3 there are actually very few Tom⁺ (thus few or zero Tom⁺/Fos⁺ cells) And while we agree the representative images shown for CA1/CA3 do not stand out, we just

couldn't get super strong signals in this area of brain with the c-Fos antibodies and Alexa fluor 647-conjugated secondary, the data was quantifiable though and so we prompt the reviewer to focus more attention on the scatter plots.

5. In fig4, were the layer specific differences observed?

Reviewer wants to know if there are layer specific differences in the auditory cortex. While we did not specifically investigate for layer differences, no obvious differences were noted during data collection. Since such a layer effect was not apparent, we do not think it is needed as it will not have any meaningful effect on the outcome of our study.

6. In fig4, Tom+ image for EphB2-/- shows low number of red cells, which contradicts results presented in graphs. cFos cells are also hard to see in blue channel.

Reviewer thinks the image in Figure 4 of Tom⁺ neurons in EphB2 mutant has low numbers of cells. We do agree that the representative image here we originally selected is not ideal and so have replaced it with another image in the revised manuscript.

Reviewer #2 (Remarks to the Author):

EphrinB receptors and their cognate ligand partners form neuronal signalling complexes linked via tyrosine kinases that act in forward as well retrograde directions. These complexes have been found to have distinct roles in development as well as synaptic function at mature synapses. In this study, the authors study EphB2 receptor signalling, and its role in fear learning, using a range of different transgenic animals. They use a classical fear learning paradigm, contextual and cued (sound) fear to study learning and memory in mice in which the EphB2 receptors have been genetically deleted, or their forward signalling has been disrupted. They show firstly that in mice in which EphB2 tyrosine kinase signalling has been disrupted, fear conditioning to both context and tone is greatly reduced. Using anatomical analysis, they show that in animals where EphB2 signalling is disrupted there are differences in the anatomical changes in identified neurons.

The major conclusion of this study is: "We show here that EphB2 receptor forward signaling is necessary for FC-induced learning and memory, with both its intracellular tyrosine kinase catalytic activity and ability to couple to PDZ domain containing proteins being particularly important for sound-cued, hippocampal-independent memories".

Overall, the experiments are well done, and the data carefully addressed. It is clear that in when EphB2 signalling is disrupted, animals have a deficit in fear learning. This effect is replicated in animals where forward tyrosine kinase signalling is specifically disrupted. These behavioural changes are mimicked by morphological changes that are seen following fear learning. Moreover, in a strain where forward tyrosine kinase signalling is enhanced there may be an enhancement in learning.

This is a very extensive data set. However, I cannot see how these data lead to the conclusions that are reached. This study clearly shows that when EphB2 signalling is disrupted, animals do not learn well. Moreover, there are clearly morphological changes that are identified in neurons that are likely engaged during learning and/or retrieval using an immediate early gene marker. However, the link between these two sets of observations is not shown to be causal.

All experiments were done comparing WT mice with those that harbour one or several mutated genes. Thus, it seems equally likely that EphB2 signalling is not in fact engaged during learning or retrieval, but, as these are transgenic strains, the lack of EphB2 signalling leads to developmental changes that block fear learning and molecular events that follow it. With regard to the Fos+ labelling, they themselves suggest that they cannot rule out the possibility that this is simply labelling sensory activated neurons. The impact of disrupting EphB2 function on synaptic or neural function are never explored. For instance, they assume that the morphological changes they describe are due to changes in synaptic plasticity - however, this is not explored for the cells in question.

Reviewer has concerns about causality and our ability to directly connect the poor LTM of EphB2 mutant mice with the observed dendritic/ synaptic morphological abnormalities. This would be a very difficult if not impossible issue to directly address with the tools we have available. Further, the Editors letter stated that demonstrating causality is not a requirement, though we have revised the discussion and acknowledge this issue in the revised manuscript.

Other points:

For the neuron counts in the method it is stated that "For neuron counts in each region of the brain assessed, the average number of Tom+ single-positive cells, Fos+ single positive cells, and Tom+/Fos+ double-positive cells that were detected in the WT home cage control mice was set to 100%".

I found this somewhat confusing. When looking at the figures (eg Fig 2, upper blade), in the first panel counts are presented for TOM+ cells for wild-type in the home cage (WT-h - labelled #1), wild type that underwent fear conditioning (WT - #2), and EphB2-/- (#3). For the WT-h animals I expected this to be a 100% - however, as can be seen the points are scattered around the 100%. And similarly for the Fos+ single positive and the Tom+/Fos+ counts.

Reviewer seems to be confused by our use of scatter plots, which we implement throughout our study/manuscript in attempts to present the data in as transparent a way as possible. In such scatter plots, the means \pm s.e.m. of each data set are indicated along with all the individual data points, and the WT-h means are set to 100% to allow for comparison to the results with the CS-US trained WT and EphB2 mutants.

Reviewers' comments:

Reviewer #1 (Remarks to the Author):

In the revised version authors addressed some of the reviewer's concerns but several major concerns are still remaining and should be addressed.

1. In fig 1 authors use pre-tone freezing as a control, but another CS only control is commonly used to make a distinction between fear conditioning effects vs effects of multiple exposure/habituation to the context/tone. Use of pre-tone freezing instead of CS control is probably acceptable here, as a habituation of "CS only" animals to context and tone would be anticipated over time (unless there is an effect of EPHB2 deletion). However, it is not acceptable to express data as a percentage of total observation times to get rid of differences in the length of the analysis interval, since fear responses show distinct temporal dynamics over the course of the exposure.

2. If authors cannot show cFos staining in the auditory cortex of CS only controls in fig 2-5, the interpretation and discussion should reflect the possibility that tone-evoked responses may be altered in these mice and contribute to the observed changes in addition to fear conditioning. So you would see reduced cFos levels in mutant auditory cortex after exposure to tone without fear conditioning, which cannot be disproved here as CS only control is missing.

3. New results on female/male differences in F620D mutant included in suppl fig2 are interesting. Authors forgot to mention it in the results section.

4. I respectfully disagree with author's interpretation of long-term memory. If authors would like to emphasize "long-term fear memory" they should compare both recalls 2-4 h after training (STM) and then again 7 days after conditioning (LTM). STM and LTM occur in parallel and the result that they see is most likely a combination of both events. I am not suggesting repeating all experiments, but to tune down their interpretation and to use terms "tone recall" and "context recall" instead of LTM. The terminology "experience-driven LTM" (page 4 line 80) is also unsuitable. Multiple processes may result in a decreased fear conditioning in EphB2 mutant, such as habituation, desensitization, deconsolidation, and extinction/relearning (Riebe et al. 2012; Singewald et al. 2015).

5. cFos staining is impossible to see in fig 3-5. The reviewer is not suggesting to change images to green channel, but to show the same image in black and white (the way it is collected) so cFos staining can be evaluated.

6. Training session 1 and 2 should be added to existing panel B in fig 1 (which has plenty of empty space) instead of making new supplemental figure.

Reviewer #1 (Remarks to the Author):

In the revised version authors addressed some of the reviewer's concerns but several major concerns are still remaining and should be addressed.

1. In fig 1 authors use pre-tone freezing as a control, but another CS only control is commonly used to make a distinction between fear conditioning effects vs effects of multiple exposure/habituation to the context/tone. Use of pre-tone freezing instead of CS control is probably acceptable here, as a habituation of "CS only" animals to context and tone would be anticipated over time (unless there is an effect of EPHB2 deletion). However, it is not acceptable to express data as a percentage of total observation times to get rid of differences in the length of the analysis interval, since fear responses show distinct temporal dynamics over the course of the exposure.

While reviewer #1 comments that "Use of pre-tone freezing instead of CS control is probably acceptable here,.....", he/she also says "However, it is not acceptable to express data as a percentage of total observation times to get rid of differences in the length of the analysis interval, since fear responses show distinct temporal dynamics over the course of the exposure." We are not exactly sure what reviewer's point is or what he/she wants us to do to alleviate their concern? Our data is obtained using standard fear conditioning and simply presents how much time each mouse spent freezing in the chamber during the testing/recall when presented to the context for five minutes (Fig 1c) or to the sound-cue for three minutes (Fig. 1d). We especially do not understand what reviewer means by stating "to get rid of differences in the length of the analysis interval" as all mice subjected to the training/tests were treated exactly the same and so there are no differences in length of analysis between any of the mice in our study. Further, Reviewer #1 did not make this type of comment during the initial review, so we are just not sure what the point here precisely is or what Reviewer wants us to do? In the end, we do not think making this data more detailed and/or complicated will provide any new or meaningful information, but rather would only serve to confuse the reader at the end of the day to the main message of this particular figure that loss of EphB function is detrimental to memory whereas gain of EphB2 function is beneficial.

2. If authors cannot show cFos staining in the auditory cortex of CS only controls in fig 2-5, the interpretation and discussion should reflect the possibility that tone-evoked responses may be altered in these mice and contribute to the observed changes in addition to fear conditioning. So you would see reduced cFos levels in mutant auditory cortex after exposure to tone without fear conditioning, which cannot be disproved here as CS only control is missing.

Done, please see revised Discussion where we acknowledge this concern.

3. New results on female/male differences in F620D mutant included in suppl fig2 are interesting. Authors forgot to mention it in the results section.

Done, please see revised Results where we now mention this detail of the data.

4. I respectfully disagree with author's interpretation of long-term memory. If authors would like to emphasize "long-term fear memory" they should compare both recalls 2-4 h after training (STM) and then again 7 days after conditioning (LTM). STM and LTM occur in parallel and the result that they see is most likely a combination of both events. I am not suggesting repeating all experiments, but to tune down their interpretation and to use terms "tone recall" and "context recall" instead of LTM. The terminology "experience-driven LTM" (page 4 line 80) is also unsuitable. Multiple processes may result in a decreased fear conditioning in EphB2 mutant, such as habituation, desensitization, deconsolidation, and extinction/relearning (Riebe et al. 2012; Singewald et al. 2015).

Done, please see revised text, we now eliminate any mention of LTM in the manuscript.

5. cFos staining is impossible to see in fig 3-5. The reviewer is not suggesting to change images to green channel, but to show the same image in black and white (the way it is collected) so cFos staining can be evaluated.

Done, please see revised figures as we now show cFos staining in black and white.

6. Training session 1 and 2 should be added to existing panel B in fig 1 (which has plenty of empty space) instead of making new supplemental figure.

Done, please see revised fig 1b.